# RIF1 promotes replication fork protection and efficient restart to maintain genome stability

Chirantani Mukherjee[1], Vivek Tripathi[1], Eleni Maria Manolika[1], Anne Margriet Heijink[2,5], Giulia Ricci[1], Sarra Merzouk[3], H. Rudolf de Boer [2], Jeroen Demmers[4], Marcel A.T.M. van Vugt[2] & Arnab Ray Chaudhuri[1]

Homologous recombination (HR) and Fanconi Anemia (FA) pathway proteins in addition to their DNA repair functions, limit nuclease-mediated processing of stalled replication forks. However, the mechanism by which replication fork degradation results in genome instability is poorly understood. Here, we identify RIF1, a non-homologous end joining (NHEJ) factor, to be enriched at stalled replication forks. *Rif1* knockout cells are proficient for recombination, but displayed degradation of reversed forks, which depends on DNA2 nuclease activity. Notably, RIF1-mediated protection of replication forks is independent of its function in NHEJ, but depends on its interaction with Protein Phosphatase 1. RIF1 deficiency delays fork restart and results in exposure of under-replicated DNA, which is the precursor of subsequent genomic instability. Our data implicate RIF1 to be an essential factor for replication fork protection, and uncover the mechanisms by which unprotected DNA replication forks can lead to genome instability in recombination-proficient conditions.

[1] Department of Molecular Genetics, Erasmus University Medical Center, Wytemaweg 80, Rotterdam 3015CN, The Netherlands. [2] Department of Medical Oncology, University Medical Center Groningen, University of Groningen, Hanzeplein 1, 9713 GZ Groningen, The Netherlands. [3] Department of Developmental Biology, Erasmus University Medical Center, Wytemaweg 80, Rotterdam 3015CN, The Netherlands. [4] Department of Biochemistry, Erasmus University Medical Center, Wytemaweg 80, Rotterdam 3015CN, The Netherlands. [5] Present address: Lunenfeld-Tanenbaum Research Institute, Mount Sinai Hospital, Toronto, ON M5G 1X5, Canada. Correspondence and requests for materials should be addressed to A.R.C. (email: a.raychaudhuri@erasmusmc.nl)

Proteins involved in the HR and FA pathways like BRCA1/2 and FANCD2 have been associated with repair of replication-associated DNA damage[1,2]. Additionally, HR and FA factors protect DNA replication forks from extensive MRE11 nuclease-mediated degradation, preventing genome instability[3,4]. This function is clinically relevant as fork protection was found to induce chemoresistance in BRCA2-defective cells[5,6]. Another parallel pathway in the processing of stalled replication forks has been identified, involving the DNA2 nuclease[7,8].

Recently, replication fork reversal was shown to be required for effective fork degradation in BRCA2-deficient cells, with the "regressed arm" being the access point for MRE11-mediated processing[9–12]. Although fork reversal is a stabilizing structure for stalled replication forks[13–16], degradation of regressed forks results in genome instability[9–12]. However, the mechanisms that regulate fork degradation-mediated genome instability remain poorly understood.

Mammalian Rap1-interacting factor 1 (RIF1) has multiple functions, including mediating NHEJ at double strand breaks (DSBs), regulation of replication origin timing, and resolution of catenanes[17–25]. In the process of DSB repair via NHEJ, RIF1 is a crucial interactor of 53BP1[17,19–21,25,26] and interacts with the N-terminal SQ/TQ sites of 53BP1[26]. Loss of RIF1 also results in resistance to PARP inhibitor treatment signifying its clinical relevance[17,20,21].

RIF1 has also been implicated in the control of replication timing in mammalian cells[18], mediated through its interaction with Protein Phosphatase 1 (PP1)[22,24]. Interestingly, Rif1-deficient mice are embryonic lethal, suggesting that RIF1 could be involved in the tolerance of high levels of replicative stress encountered during proliferation of stem cells[27,28].

Here, we show a novel role for RIF1 in the protection of reversed replication forks from DNA2-mediated degradation. Furthermore, the C-terminal domain of RIF1—responsible for binding both protein phosphatase 1 as well as cruciform DNA structures—is essential for protecting reversed forks from degradation. Finally, we provide evidence that degradation of reversed forks is linked to defective replication restart in RIF1-deficient cells, resulting in the accumulation of under-replicated DNA and subsequent genome instability.

## Results

**RIF1 is recruited to stalled DNA replication forks**. To identify novel factors enriched at stalled replication forks, we utilized iPOND (isolation of proteins on nascent DNA) coupled with SILAC (stable isotope labeling of amino acids in cell culture)-based quantitative mass-spectrometry[29,30]. Mouse embryonic stem cells were treated with hydroxyurea (HU) to stall DNA replication forks and subsequently subjected to quantitative mass-spectrometry to analyze the proteomes associated with the replication forks (Fig. 1a and Supplementary Data 1). Seven-hundred twenty-one proteins were identified commonly between two independent experiments (Supplementary Fig. 1a). We identified RIF1 among 44 proteins, which showed >2-fold enrichment upon HU treatment (Fig. 1b and Supplementary Data 1). Consistent with previous reports, we also observed over two-fold increase in replication stress response proteins, including RAD51 and RPA2 (Fig. 1c and Supplementary Data 1)[29,30]. Whereas core components of the replicative helicase, including MCM2-7, largely remained unchanged at time of early replication stress, PCNA enrichment decreased at stalled replication forks, as reported previously[30] (Fig. 1c).

To further verify the recruitment of RIF1 to stalled forks, we performed immunofluorescence analysis to measure localization of RIF1 at sites of DNA replication. Wild type (WT) mouse embryonic fibroblasts (MEFs) were incubated with EdU, and localization of RIF1 to sites of EdU incorporation was measured in the presence or absence of HU treatment (Fig. 1d). Approximately 50% of the WT cells in non-treated condition (NT) showed EdU incorporation, (Fig. 1d, e). Only a small fraction of EdU-positive WT cells in non-treated cells were positive for RIF1 foci. By contrast, upon HU treatment, approximately 80% of the EdU-positive cells showed EdU co-localization with RIF1 (Fig. 1d, e and Supplementary Fig. 1c). To verify that EdU and RIF1 co-localization upon HU treatment indeed occurred at stalled forks, we performed proximity ligation-based assays (PLA) to detect RIF1 binding to replicated DNA[12]. WT cells treated with HU displayed a significant increase in PLA signals per cell. However, the total percentage of PLA-positive cells (signifying the replicating population) did not increase significantly (Supplementary Fig. 1d), suggesting that RIF1 is recruited to stalled DNA replication forks.

Since RIF1 localization to sites of DSBs depends on 53BP1[17,19–21,25,26] (Supplementary Fig. 1b, e), we tested if localization of RIF1 to stalled replication forks also required 53BP1. Interestingly, upon HU treatment 53bp1−/− MEFs showed similar levels of RIF1-EdU co-localization as WT cells (Fig. 1d, e and Supplementary Fig. 1b, c). Finally, we tested whether 53BP1 also localized to sites of DNA stalled forks upon HU treatments. In WT MEFs we observed 53BP1 foci in a low percentage of cells, but these foci did not co-localize with EdU (Supplementary Fig. 1f). Furthermore, HU treatments did not significantly increase the percentage of 53BP1-positive cells (Supplementary Fig. 1f), suggesting that RIF1 is enriched at stalled replication forks, independently of 53BP1.

**RIF1 protects reversed DNA replication forks**. To explore the role of RIF1 during unperturbed DNA replication, we monitored the frequency of replicating cells by incorporation of EdU by flow cytometry (Supplementary Fig. 2a–d). WT and Rif1−/− cells showed similar percentages and intensities of EdU staining (Supplementary Fig. 2b–d). Additionally, we analyzed progression rates of individual replication forks in WT and Rif1−/− cells by DNA fiber assay. We sequentially labeled cells with CldU (red) and IdU (green), followed by tract length analysis (Fig. 2a and Supplementary Fig. 7a). WT and Rif1−/− cells revealed no significant difference in tract lengths, again suggesting that RIF1 is not essential for unperturbed DNA replication (Fig. 2a).

Next, we tested if RIF1 was involved in stabilizing DNA replication forks under stressed conditions. WT and Rif1−/− MEFs were sequentially labeled with CldU and IdU. On-going replication forks were then stalled with HU (Fig. 2b). The relative shortening of the IdU tract after HU treatment served as a measure of fork degradation (Fig. 2b). Upon HU treatment, WT cells showed tract lengths similar to non-treated cells with mean ratio close to 1 (Fig. 2b). Contrastingly, RIF1-deficient cells displayed a significant reduction in the IdU tract lengths (Fig. 2b and Supplementary Fig. 7b). Human RIF1 knock-out HAP1 cells (RIF1-KO)[23], also revealed a similar trend as observed in Rif1−/− MEFs (Fig. 2c and Supplementary Figs. 2e and 7c). This suggests that RIF1 is essential for protection of replication forks from degradation (Fig. 2b, c). Recently, 53BP1 deficiency in B-lymphocytes was demonstrated to cause degradation of nascent strands[31]. Since RIF1 interacts with 53BP1 for NHEJ, we tested whether protection of nascent strands by RIF1 could be dependent on this interaction. Analysis of 2 different clones of 53bp1−/−MEFs did not show fork degradation upon HU treatment. This suggests that in our experimental setup, the role of RIF1 in replication fork protection is independent of 53BP1 (Fig. 2d and Supplementary Fig. 7d).

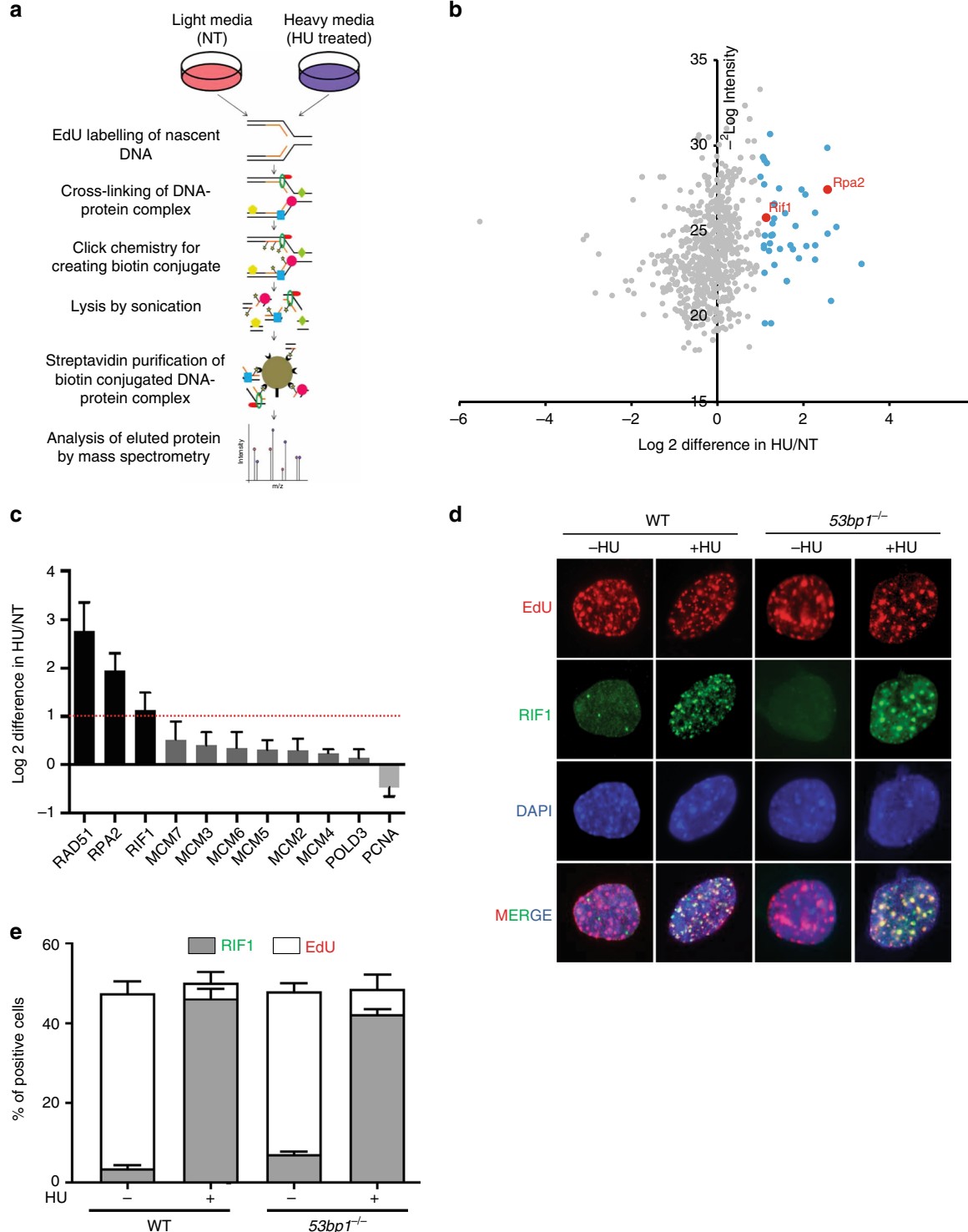

**Fig. 1** RIF1 is recruited to the stalled replication forks. **a** Schematic representation of iPOND experiment. **b** Volcano plot showing the results for average fold-change to identify significantly upregulated proteins upon HU treatment based on H:L ratio in the SILAC experiment. The *x*-axis ('2Log Difference HU/NT) represents the fold upregulation. Data points in blue represent proteins that are upregulated >2-fold; RIF1 is indicated in red. **c** Bar graph showing fold upregulation of a selection of proteins upon HU treatment based on their SILAC H:L ratios (error bars represent standard deviation). **d** Representative micrographs showing co-localization of RIF1 (green) to sites of DNA replication as marked by EdU (red) in the presence or absence of HU in WT and *53bp1*[−/−] cells. Nucleus was stained with DAPI (blue). **e** Quantitation of **d** showing the percentage of cells, which show co-localization between EdU and RIF1(error bars represent standard deviation)

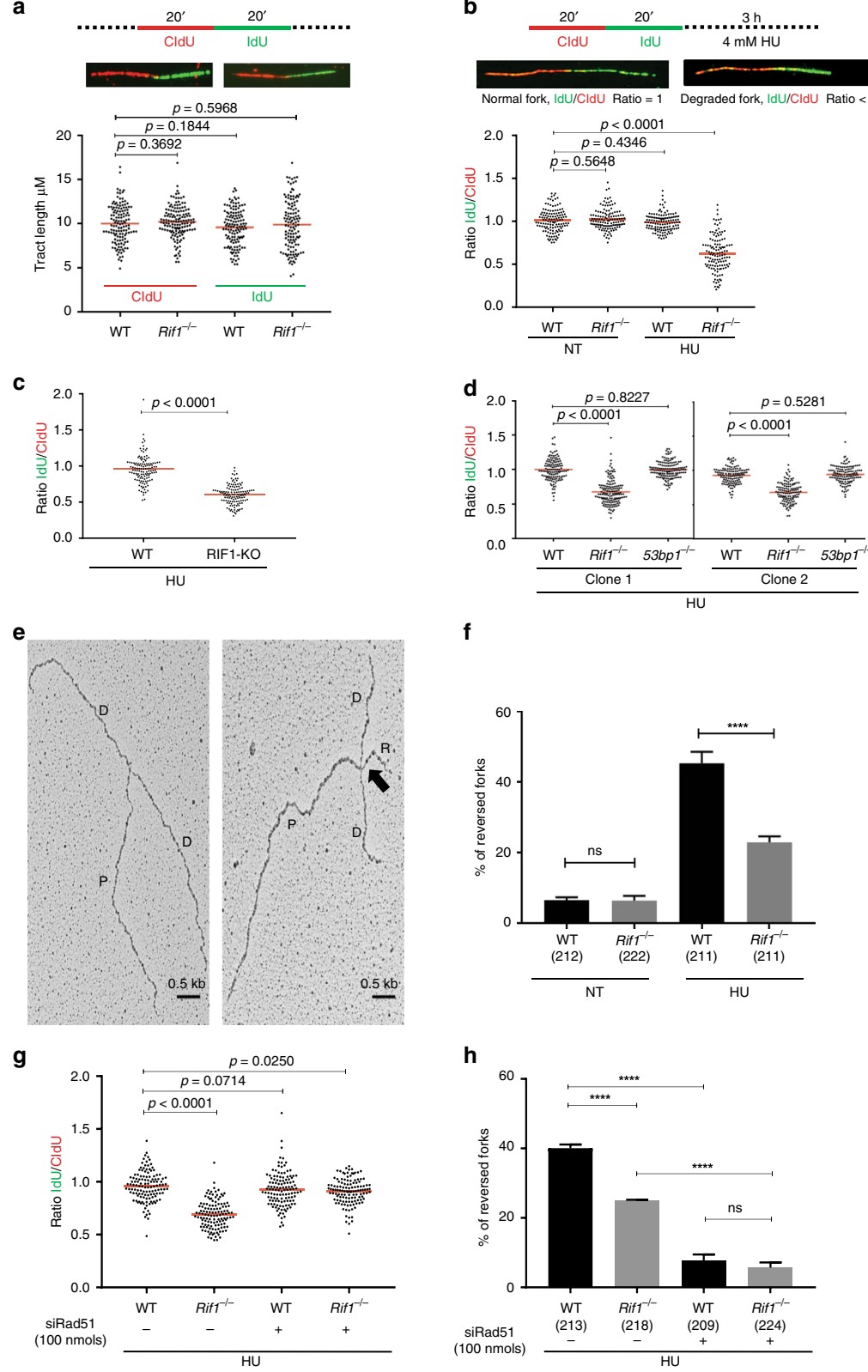

Fork degradation has been associated with loss of HR factors[3,4]. We, therefore, tested if loss of RIF1 also resulted in HR defects. Localization of the RAD51 recombinase to sites of DNA DSBs has been shown to be a reliable readout for functional HR[32]. Upon ionizing irradiation, Rif1[−/−] MEFs were proficient in forming RAD51 foci (Supplementary Fig. 3a, b). Additionally,

we monitored HR efficiency using the DR-GFP reporter[33]. Consistent with earlier reports[20,34,35], Rif1[−/−] MEFs did not show a significant difference in HR frequencies when compared to WT cells (Supplementary Fig. 3c). Finally, we tested the ability of RIF1-deficient cells to form sister chromatid exchanges (SCEs) in the presence or absence of HU or cisplatin. Treatments with

**Fig. 2** Protection of reversed forks from degradation by RIF1. **a** Top panel: schematics of experimental conditions for fork progression in WT and $Rif1^{-/-}$ MEFs. Cells were labeled with CldU (red) followed by IdU (green) as indicated. Representative DNA fibers for progression in WT and $Rif1^{-/-}$ MEFs are shown below the schematic. Progression was measured by tract lengths of CldU (red) and IdU (green) in micrometers (µM). **b** Top panel: schematic for labeling cells in fork degradation assay. Representative pictures of normal and degraded fork are shown below the schematic. Cells were labeled with CldU followed by IdU and then subjected to replication stress with 4 mM HU for 3 h. Ratio of IdU to CldU tract length was plotted as readout for fork degradation. **c, d** Fork degradation assay in WT and RIF1-KO HAP1 cells (**c**) and between two different clones of WT, $Rif1^{-/-}$, and $53bp1^{-/-}$ MEF cell line (**d**). Experimental conditions were similar as in **b**. **e** Representative electron micrographs of normal fork (left) and reversed replication fork (right) observed on treatment with HU. The black arrow pointing to four-way junction at the replication fork indicates fork reversal (P, Parental, D, Daughter strand, R, Reversed arm). **f** Percentage of fork reversal in WT and $Rif1^{-/-}$ MEFs treated with or without HU (4 mM) for 3 h. Numbers of analyzed molecules are indicated in parentheses. **g** WT and $Rif1^{-/-}$ MEFs were transfected with siRad51 (100 nmols, 48 h) followed by labeling and treatment with 4 mM HU for 3 h. Fork degradation was determined in the presence and absence of RAD51. **h** Fork reversal frequencies observed with and without depletion of RAD51 in WT and $Rif1^{-/-}$ MEFs under HU treatment. Numbers of analyzed molecules are indicated within parenthesis. Red bars in **a, b, c, d,** and **g** represent mean values from 125 fibers from each genotype under each condition. $P$-values were derived from Kruskal–Wallis ANOVA with Benjamini Hochberg (BH) post test except in **c**, where Mann–Whitney was used and in **f** and **h**, where unpaired $t$-test was done (ns, non-significant, ****$P < 0.0001$). All experiments were repeated three times with similar outcomes (Supplementary Data 2 and Supplementary Fig. 7a–e)

either HU or cisplatin significantly increased the number of SCEs in both WT and $Rif1^{-/-}$ cells. However, no significant differences in SCEs were observed between WT and $Rif1^{-/-}$ cells (Supplementary Fig. 3d–e). Taken together, these data suggest that loss of RIF1 does not result in defective HR.

DNA replication stress results in fork reversal[14]. Recent reports have identified reversed forks to be the substrate for nascent stand degradation in the absence of BRCA2[9–12]. We, therefore, hypothesized that RIF1 -like BRCA2- could be involved in the protection of reversed forks. To assess replication fork architecture in WT and $Rif1^{-/-}$ cells, we visualized replication intermediates formed in vivo using electron microscopy (EM)[36]. HU treatment of WT MEFs resulted in a high percentage of reversed replication forks (Fig. 2e, f and Supplementary Data 3). In contrast, HU-treated $RIF1$-deficient cells showed a significantly lower frequency of fork reversal (Fig. 2f and Supplementary Data 3). These data suggest that RIF1 could either be involved in mediating fork reversal or in protecting reversed forks.

RAD51 has been shown to be essential for mediating fork reversal[11,37,38]. RAD51 downregulation rescues fork degradation in BRCA2-deficient cells, suggesting that unprotected reversed forks are the substrates for degradation[10,11,39]. However, stabilization of RAD51 on the reversed forks is also important for protection of reversed forks[40]. To test if RIF1 is involved in fork reversal, we downregulated RAD51 in WT and RIF1-deficient MEFs and tested for fork degradation (Fig. 2g and Supplementary Fig. 3f). Near-complete downregulation of RAD51 in WT cells did not induce fork degradation in WT cells (Fig. 2g)[10,11,39]. However, RAD51 downregulation in $Rif1^{-/-}$ cells significantly rescued fork degradation, suggesting that RIF1 is required for fork protection but not for reversal of forks (Fig. 2g and Supplementary Fig. 3f, 7e). Consistently, our EM analysis showed that knockdown of RAD51 in WT cells resulted in almost complete abolishment of fork reversal upon HU treatments (Fig. 2h and Supplementary Data 3)[11,37,38]. However, this decrease in fork reversal was not further affected by RIF1 inactivation (Fig. 2h and Supplementary Data 3). To subsequently test if RIF1 acts epistatic to RAD51 in protecting reversed forks, we partially downregulated RAD51 in WT and $Rif1^{-/-}$ cells and assessed fork degradation (Supplementary Figs. 3g, h and 7f). Partial downregulation of RAD51 resulted in fork degradation in WT cells, but did not result in aggravated degradation observed upon RIF1-deficiency alone, suggesting that RIF1 could also be involved in the stabilization of RAD51 on the reversed arm (Supplementary Figs. 3h and 7f). Taken together, these data strongly suggest that RAD51 acts upstream of RIF1 in fork reversal and that RIF1 could be involved in the protection of reversed forks, rather than the process of fork reversal itself.

**Fork degradation in RIF1-deficient cells mediated by DNA2.** Since MRE11 has been implicated in mediating replication fork degradation[3,4], we tested if MRE11 is also responsible for fork degradation upon RIF1- deficiency. We downregulated MRE11 in WT and $Rif1^{-/-}$ MEFs (Fig. 3a) and measured fork degradation. Downregulation of MRE11 in RIF1-deficient cells resulted in a partial but significant rescue of fork degradation (Fig. 3b and Supplementary Fig. 7g). Since partial rescue of fork degradation could result from residual MRE11 activity, we treated cells with the MRE11 inhibitor Mirin[41]. Mirin treatment failed to completely rescue the fork degradation phenotype in $Rif1^{-/-}$ MEFs, again suggesting that MRE11 is not the main nuclease involved in degradation of replication forks in $Rif1^{-/-}$ cells (Supplementary Fig. 4a and 7h). DNA2 nuclease has been implicated in the restart of reversed replication forks[37] and the uncontrolled degradation of stalled replication forks[7,8]. Therefore, we tested if DNA2 was involved in the degradation of replication forks in $Rif1^{-/-}$ MEFs. Downregulation of DNA2 completely rescued the fork degradation in $Rif1^{-/-}$ MEFs (Fig. 3a, b). We next analyzed the involvement of DNA2 in fork degradation in RIF1-KO HAP1 cells, using the DNA2 inhibitor NSC-105808 (DNA2i)[42]. Pretreatment of RIF1-KO cells with DNA2i significantly rescued the degradation of nascent strands (Fig. 3c), and no additional rescue was observed upon combined Mirin and DNA2i treatments (Fig. 3c and Supplementary Fig. 7i). A dependency on DNA2 for fork degradation was also confirmed in $Rif1^{-/-}$ MEFs, using either Mirin, DNA2i or both (Supplementary Figs. 4a and 7h). To verify the context specificity for DNA2, we pretreated $Brca1^{-/-}$ MEFs with either Mirin, DNA2i or both and assessed the rescue of fork degradation. While Mirin treatment rescued fork degradation as expected (Supplementary Figs. 4b and 7j), DNA2i treatment only partially rescued fork degradation in $Brca1^{-/-}$ cells. Additionally, combined inhibition of MRE11 and DNA2 in $Brca1^{-/-}$ cells did not show any additive effect (Supplementary Figs. 4b and 7j), suggesting that DNA2 is the main nuclease driving fork degradation in RIF1-deficient cells.

Next, to test if DNA2 inhibition could rescue formation of reversed forks upon RIF1-deficiency, cells were treated with HU in the presence or absence of DNA2i, and the frequencies of reversed forks were analyzed. As observed earlier, $Rif1^{-/-}$ MEFs treated with HU displayed a significantly reduced frequencies of reversed forks. Strikingly, treatment with DNA2i and HU in $Rif1^{-/-}$ cells significantly rescued fork reversal (Fig. 3d and

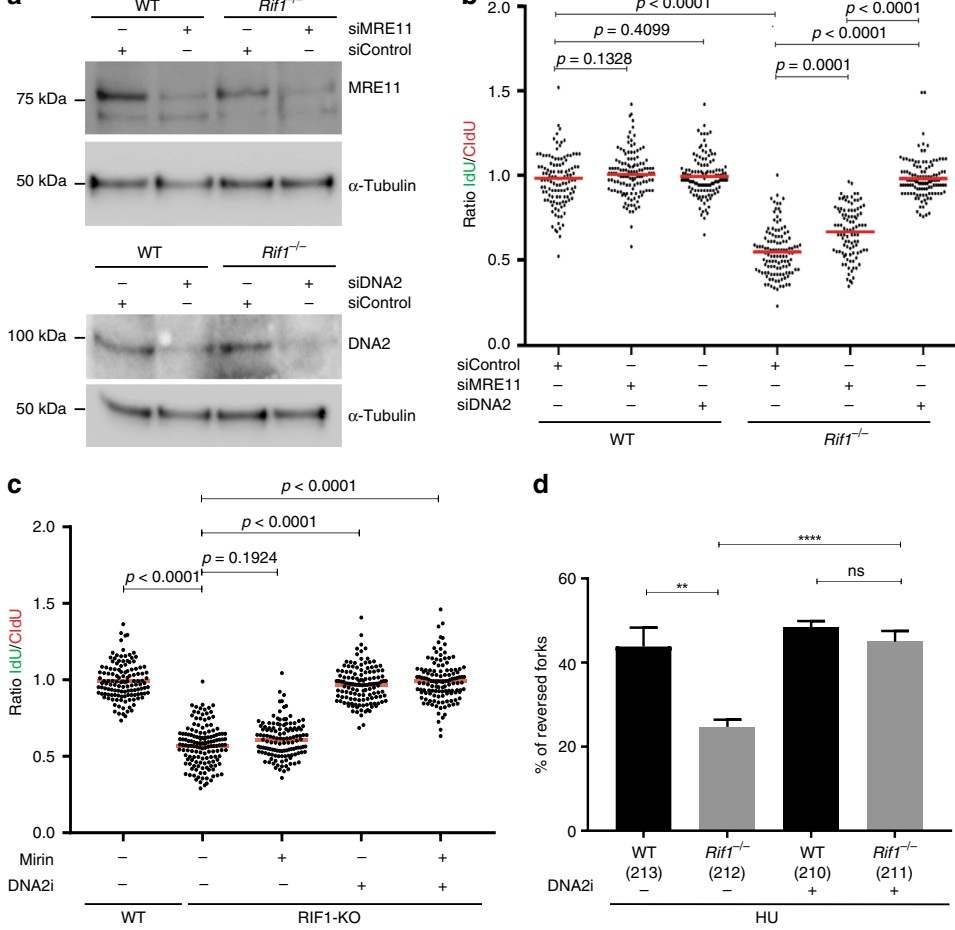

**Fig. 3** DNA2 drives reversed fork degradation in RIF1-deficient cells. **a** Western blot analysis for the downregulation of MRE11 and DNA2 in WT and $Rif1^{-/-}$ MEFs. WT and $Rif1^{-/-}$ MEFs were transfected with either siControl or siRNAs smart pool against MRE11 and DNA2. Lysates made were probed with antibody against MRE11 and DNA2. Tubulin is used as loading control. **b** Ratio of IdU versus CldU in WT and $Rif1^{-/-}$ MEFs upon HU treatment after downregulating Mre11 or DNA2 (**a**). **c** Ratio of IdU versus CldU in WT and RIF1-KO HAP1 cells upon HU treatment after inhibiting Mre11 and DNA2 using mirin and DNA2 inhibitor. **d** Electron microscopic analysis of percentage of reversed forks observed in WT and $Rif1^{-/-}$ MEFs subjected to HU (4 mM) for 3 h, with or without DNA2 inhibitor. Numbers of analyzed molecules are indicated in parentheses. At least 125 readings were taken for **b** and **c** and the mean ratio is represented by red bar. $P$-values were derived from Kruskal–Wallis ANOVA with Benjamini Hochberg post test except in **d**, where unpaired $t$-test (ns, non-significant, ****$P < 0.0001$, **$P = 0.0024$) was carried out. Similar observation was made from three independent experiments (Supplementary Data 2 and Supplementary Fig. 7g, i)

Supplementary Data 3). Altogether, these data show that RIF1 is responsible for protecting reversed replication forks from DNA2-mediated degradation.

**C-terminal region of RIF1 is essential for fork protection**. Mammalian RIF1 has two conserved regions at its termini[20]. The N-terminus consists of HEAT-like α-helical repeats (HEAT-repeats) and is required for Rif1 recruitment to sites of DSBs[20]. The C-terminal domain (CTD) of RIF1 consists of three sub-domains (CI, CII and CIII) and confers in vitro DNA binding activity, preferentially to cruciform structures[43]. Mammalian RIF1 also contains two PP1 interaction motifs, which are responsible for the control of replication timing[24,40,43].

To test which domain of RIF1 is responsible for protection of reversed replication forks, we generated truncation constructs from a human full-length RIF1 construct (hRIF1-FL)[20]. Deletion constructs were generated for the HEAT domain (Del-HEAT), CTD domain (Del-CTD), CI domain (Del-CI), and CII domain (Del-CII) (Fig. 4a). These constructs were then transfected into $Rif1^{-/-}$ MEFs and checked for their expression levels (Fig. 4b).

Complementation with hRIF1-FL and Del-HEAT significantly rescued the fork degradation observed in $Rif1^{-/-}$ MEFs (Fig. 4c). In contrast, expression of RIF1 deletion mutants with either the CI, CII domains or the whole CTD failed to rescue the fork degradation in RIF1-deficient cells (Fig. 4c and Supplementary Fig. 8a). Furthermore, complementation of $Rif1^{-/-}$ MEFs with either hRIF1-FL or Del-HEAT constructs resulted in a ~2-fold increased fork reversal frequency when compared with $Rif1^{-/-}$ MEFs (Fig. 4d and Supplementary Data 3). In contrast, $Rif1^{-/-}$ MEFs with either Del-CI or Del-CII failed to restore fork reversal frequencies in these cells (Fig. 4d). These data suggest that the CI and CII domains of RIF1, which contain interaction motifs for PP1 and have DNA cruciform binding properties, are essential for protection of reversed forks.

To directly test the involvement of PP1 in replication fork protection, we depleted PP1 in WT and $Rif1^{-/-}$ MEFs and assessed fork degradation (Supplementary Fig. 4c). Interestingly, depletion of PP1 in WT cells resulted in significant fork degradation upon HU treatments, which was epistatic with RIF1 (Fig. 4e and Supplementary Fig. 8b). Furthermore, pretreatment of WT and $Rif1^{-/-}$ MEFs with the selective PP1

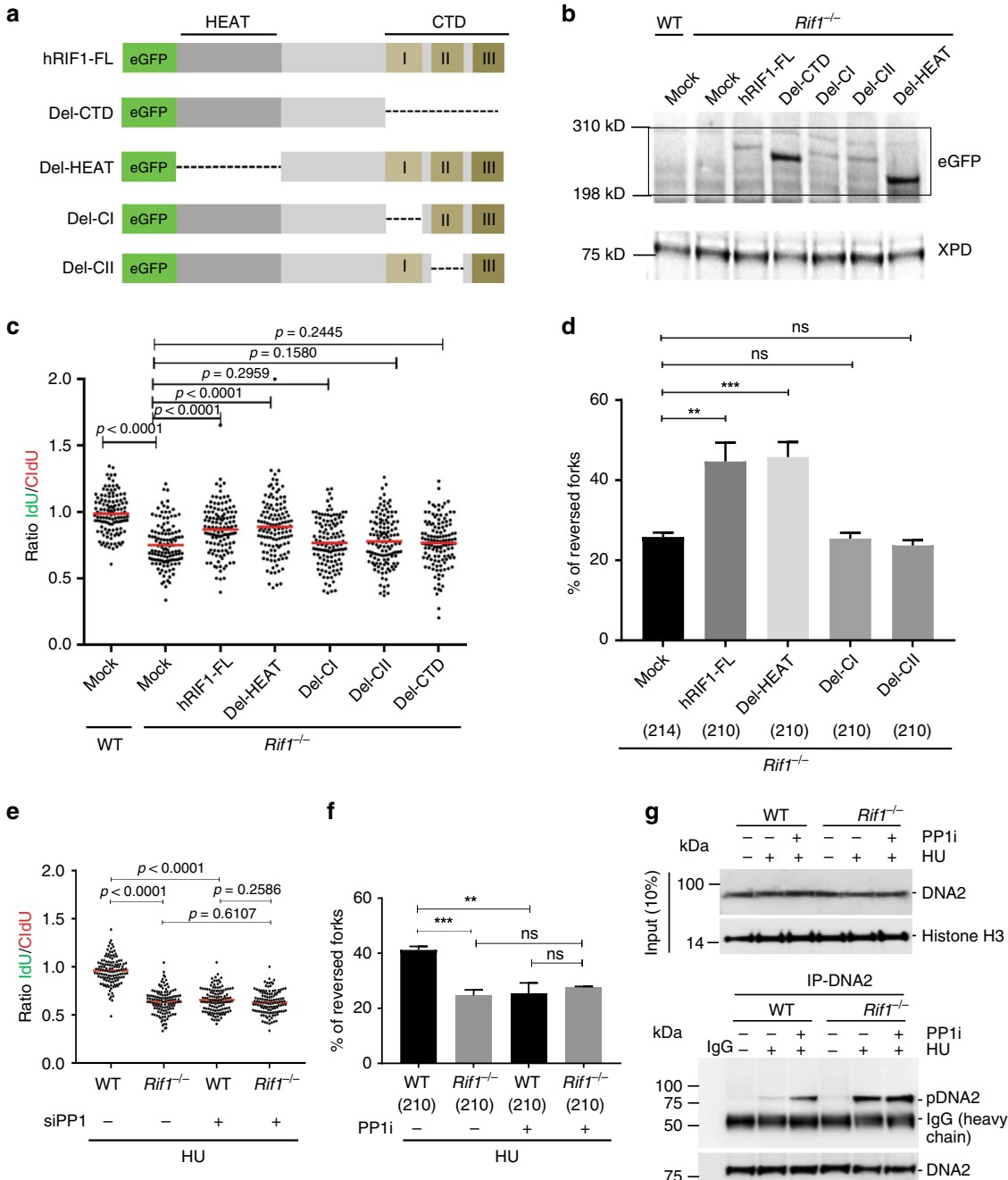

**Fig. 4** C-terminal region of RIF1 protects of reversed forks from degradation. **a** Schematic of full-length (FL) human RIF1 protein and deletion mutant constructs. Deleted region for each mutant is denoted by dotted line. **b** Western blot analysis for $Rif1^{-/-}$ MEFs transfected with mutant construct of human RIF1. Lysates were probed with antibody against GFP. XPD was used as loading control. Expression of mutant protein is visualized as distinct bands in range of 198 kD to 310 kD, which is missing in mock-transfected samples. **c** DNA fiber assay to assess the rescue of Fork degradation in $Rif1^{-/-}$ MEFs transfected with RIF1 mutant constructs (for 48 h) upon treatment with 4 mM HU for 3 h. **d** Percentage of fork reversal in $Rif1^{-/-}$ MEFs transfected with different mutant constructs of human RIF1 and subsequent treatment with HU for 3 h (4 mM). Numbers of analyzed molecules are indicated in parentheses. **e** DNA fiber assay to determine the extent of fork degradation in WT and $Rif1^{-/-}$ MEFs upon siRNA-mediated downregulation of PP1. **f** Percentage of reversed forks observed in WT and $Rif1^{-/-}$ MEFs treated with 4 mM HU for 3 h with or without inhibiting PP1. Number of molecules analyzed are indicated within the parenthesis. At least 125 readings were taken for **c** and **e** and the mean values are represented by red bar. $P$-values were derived from Kruskal–Wallis ANOVA with Benjamini Hochberg post test for **c** and **e** and from unpaired $t$-test for **d** (ns, non-significant, ***$P = 0.0009$, **$P = 0.0025$) and **f** (ns, non-significant, ***$P = 0.0003$, **$P = 0.0026$). All the experiments were repeated for three times with similar outcomes (Supplementary Data 2 and Supplementary Fig. 8a, b). **g** DNA2 is hyper phosphorylated in $Rif1^{-/-}$ MEFs during replication stress. Top panel: level of DNA2 in nuclear extracts from WT and $Rif1^{-/-}$ MEFs before and after treatment with HU alone or in combination with PP1 inhibitor treatment (tautomycetin 225 nM for 2 h). Western blots were performed with antibody against DNA2 antibody. Histone H3 was used as loading control. Bottom panel: Immunoprecipitations were carried out with anti-DNA2 antibody or the corresponding IgG and were probed with p-(S/T) antibody

inhibitor tautomycetin (PP1i)[44,45] resulted in similar degradation profiles as observed upon knockdown of PP1 (Supplementary Figs. 4d and 8c). HU treatment after PP1 inhibition in WT cells also resulted in a significant decrease of fork reversal frequencies (Fig. 4f and Supplementary Data 3). PP1i treatment in RIF1-deficient cells did not further decrease the amount of fork reversal to levels observed in either $Rif1^{-/-}$ cells or WT cells treated PP1i (Fig. 4f), suggesting that RIF1 and PP1 are epistatic for preventing the degradation of reversed forks (Fig. 4e, f, Supplementary Fig. 8b, c and Supplementary Data 3).

DNA2 phosphorylation was shown to be important for recruitment to DSBs in yeast[46]. We, therefore, hypothesized that access of DNA2 to forks upon replication stress could be controlled by PP1 in a phosphorylation-dependent manner. To test this hypothesis, we immunoprecipitated DNA2 from nuclear extracts of WT or $Rif1^{-/-}$ MEFs treated with either HU or HU and PP1i. The immunoprecipitated DNA2 was then probed for phosphorylation status using the phospho-S/TQ motif antibody. Treatment with HU slightly increased the levels of DNA2 phosphorylation in WT cells when compared to untreated cells (Fig. 4g). PP1 inhibition markedly increased the phosphorylation levels of DNA2 upon HU treatment (Fig. 4g). Additionally, DNA2 phosphorylation levels in RIF1-deficient cells upon HU treatments were observed to be similar to WT cells upon PP1 inhibition. The DNA2 phosphorylation status was not further increased upon inhibition of PP1 in RIF1-deficient cells upon HU treatment, suggesting that RIF1-PP1 interaction controls DNA2 phosphorylation levels upon replicative stress (Fig. 4g).

**RIF1 deficiency results in defective fork restart**. Nascent strand degradation has been linked to increased genome instability[4]. We, therefore, tested if fork degradation in RIF1-deficient cells induces immediate induction of DSBs. We performed pulsed-field gel electrophoresis (PFGE) analysis[16] where we did not observe a significant difference between WT and $Rif1^{-/-}$ MEFs (Fig. 5a, b). Treatment with HU resulted in a marginal but non-significant increase of DSBs both in WT and $Rif1^{-/-}$ cells, when compared to their non-treated counterparts (Fig. 5a, b). These low levels of DSBs observed in RIF1-deficient cells were not suggestive of fork collapse into DSBs upon degradation, a phenomenon that was observed on the entire population of active forks (3000–12,000 per cell)[47] (Fig. 5b).

We next tested if fork degradation resulted in genome instability in WT and $Rif1^{-/-}$ MEFs treated with replication stress-inducing agents HU, cisplatin, and Camptothecin (CPT) (Fig. 5c, d and Supplementary Fig. 5a) by metaphase spreads. Untreated $Rif1^{-/-}$ cells did not show a significant increase in aberrant chromosomes (Fig. 5c, d and Supplementary Fig. 5a). However, upon HU, cisplatin or CPT treatment, $Rif1^{-/-}$ MEFs displayed significantly increased aberrations when compared to their WT counterparts (Fig. 5c, d and Supplementary Fig. 5a). Furthermore, consistent with previous data[27,48], clonogenic survival assays performed in WT and $Rif1^{-/-}$ MEFs showed that RIF1 deficiency also resulted in increased sensitivity to HU, cisplatin or CPT (Fig. 5e, f and Supplementary Fig. 5b). These data show that although fork degradation does not result in immediate replication fork collapse, it results in increased genome instability and sensitivity to replication stress.

We hypothesized that the increased genome instability in RIF1-deficient cells could arise from defective restart of stalled replication forks. To test this, we performed a fork restart assay, in which cells were labeled with CldU followed by HU treatment to stall the forks and then released into IdU (Fig. 5g). However, WT and RIF1-deficient cells did not reveal a significant difference between stalled versus restarted forks, suggesting that the

majority of forks were restarted (Supplementary Fig. 5c). Further analysis of individual tract lengths revealed that restarted forks from $Rif1^{-/-}$ cells showed significantly shorter IdU tracts, suggestive of a delayed restart in these cells (Fig. 5h and Supplementary Fig. 8d). A similar trend of delayed fork restart was also observed in RIF1-KO HAP1 cells (Supplementary Figs. 5d and 8e). Shorter inter-origin distances could also account for smaller IdU labels in RIF1-deficient cells upon restart. To test this, we allowed the forks to restart after HU treatments for multiple time points ranging from 15′ to 60′. A significant decrease in the percentage of restarted forks was observed at early time points after release (15′ and 30′) in $Rif1^{-/-}$ cells, but not at later time points (45′ and 60′) (Supplementary Fig. 5e). However, the CldU tract lengths at 30′, 45′, and 1 h show significant shorter tracts in RIF1-deficient cells, suggesting that the shorter tracts could be due to delayed restart in these cells (Supplementary Figs. 5f and 8f).

Since 53BP1 was recently implicated in replication fork restart[49], we wondered if the restart defect observed upon RIF1 inactivation is epistatic with 53BP1. To this end, we used 53BP1[15A] MEFs, which lack 15S/TQ phosphorylation sites within 53BP1 essential for RIF1 binding[26]. 53BP1[15A] cells did not display a defect, suggesting that RIF1 and 53BP1-mediated restart is differentially regulated (Supplementary Figs. 5g and 8g). These data suggested that the genome instability and sensitivity observed in RIF1-deficient cells could be a result of defective restart in these cells.

**Restart delay results in genome instability**. To explore whether fork restart defects in RIF1-deficient cells cause genome instability, we tested directly if forks restarted after HU treatments resulted in formation of DSBs. WT and $Rif1^{-/-}$ MEFs were assayed for formation of DSBs by PFGE at 15 h after release from HU-induced fork stalling. As observed previously, HU treatment in either WT or RIF1-deficient cells did not cause a significant change in DSBs frequency (Figs. 5a, b and 6a, b). However, $Rif1^{-/-}$ cells displayed a significant increase of DSBs compared to WT cells (Fig. 6a, b), which could be a result of decreased repair of DSBs after release. Since RIF1-deficient do not have a HR defect, we also tested if these cells show defective NHEJ. Using a reporter-based NHEJ assay[50], we found a significant decrease in NHEJ levels in RIF1-deficient cells when compared to WT cells, consistent with published evidence[17] (Supplementary Fig. 6a). Since the 53BP1-RIF1 axis is responsible for NHEJ repair, we next tested if 53BP1 deficiency also resulted in genome instability upon replication stress. In contrast to RIF1-deficient cells, which showed high levels of genome instability, 53BP1 deficiency did not result in significant chromosomal aberrations upon either HU or cisplatin treatments (Supplementary Fig. 6b). Taken together, these data suggest that defective NHEJ-mediated repair could be involved in increased DSB formation in RIF1-deficient cells upon restart. However, this cannot completely account for the increased genome instability observed in RIF1-deficient cells, as $53bp1^{-/-}$ MEFs did not show increased levels of genome instability upon induction of replication stress.

To further test if the delayed restart resulted in increased single-stranded DNA (ssDNA) levels in these cells, we analyzed RPA2, a surrogate for ssDNA, by flow cytometry. Upon HU treatment, the replication-associated RPA2 signals were markedly enhanced in both $Rif1^{-/-}$ and WT cells (Supplementary Fig. 6c, d). At 5 h after release from a HU-mediated block, slightly reduced but still significantly higher levels of RPA were observed in both the cell types (Supplementary Fig. 6c, d). However, at 15 h after HU release, WT cells showed low RPA levels, along with normal cell cycle profiles. In contrast, $Rif1^{-/-}$ cells displayed an accumulation

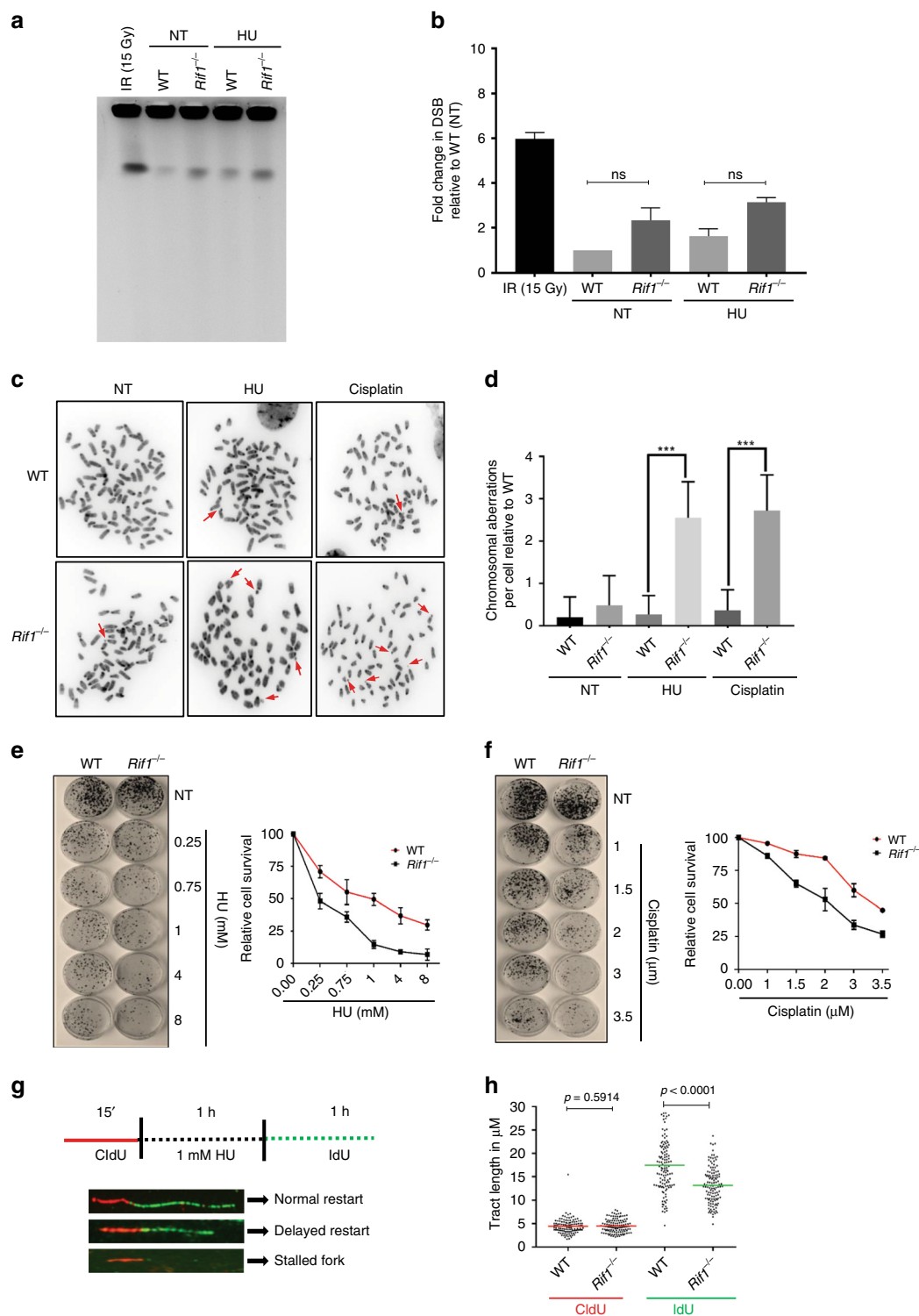

of cells in late S/G2 with significantly higher percentages of RPA2-positive cells (Supplementary Fig. 6c–e). To subsequently test if RIF1-deficient cells entered mitosis with high levels of under-replicated DNA, we performed co-staining for phospho-histone H3 in combination with RPA2 using the same experimental conditions as in Supplementary Fig. 6c. However, we did not observe any significant differences in phospho-histone H3-positive cells between the two genotypes (Supplementary Fig. 6f), suggesting that upon restart, RIF1-deficient cells expose increased

amounts of ssDNA, which causes accumulation in late S/G2 phase of the cell cycle.

We speculated that the increased levels of ssDNA in RIF1-deficient cells could be a result of under-replicated DNA during the restart process. To test this hypothesis, we performed EM analysis of restarted forks. Interestingly, we observed a significant increase in replication intermediates with high levels of ssDNA at forks in RIF1-deficient cells when compared to WT cells (Fig. 6c, e). Furthermore, a significant increase was observed in ssDNA gaps

**Fig. 5** Delayed fork restart and genomic instability observed upon RIF1 deficiency. **a** PFGE analysis for DSBs in WT and $Rif1^{-/-}$ MEFs with and without treatment with HU for 3 h. WT MEFs treated with IR (15 Gy) was taken as positive control. **b** Quantification of experiment (**a**), an integration of three independent experiments showing DSB levels relative to WT untreated (NT), (ns, not-significant, from unpaired $t$-test). **c** Representative images for analysis of genomic instability analysis by metaphase spread in WT and $Rif1^{-/-}$ MEFs upon HU and Cisplatin treatment. **d** Quantitation of chromosomal aberrations in **c**. Sixty metaphase fields per conditions were analyzed and three independent experiments were carried out. $P$-value was calculated by unpaired $t$-test (***$P \leq 0.0001$). **e–f** Images for clonogenic survival assay in WT and $Rif1^{-/-}$ MEFs treated with different concentrations of HU (**e**) and Cisplatin (**f**) after which the drugs were washed off and the cells were allowed to grow for 8 days. Adjoining graphs show the data from three independent experiments. Error bars represent s.e.m. **g** Schematics of fork restart assay by DNA fibers and representative images for normal restart, delayed restart and stalled fork upon release from HU treatment. **h** Quantitation for restart assay in **g**. Tract lengths of IdU and CldU were quantified in WT and $Rif1^{-/-}$ MEFs upon restart after treatment with 1 mM HU for 1 h from 125 fibers per sample. Red and green bars indicate mean CldU and IdU tract length. $P$-values were derived from Kruskal–Wallis ANOVA with Benjamini Hochberg post test. All experiments were repeated three times (Supplementary Data 2 and Supplementary Fig. 8d)

behind forks in $Rif1^{-/-}$ cells (Fig. 6d, f). To test if ssDNA regions observed in RIF1-deficient cells were a result of the fork degradation process, we inhibited DNA2 in WT and $Rif1^{-/-}$ cells before release from HU block. Interestingly, DNA2 inhibition in RIF1-deficient cells significantly reduced both the ssDNA regions at the forks and the gaps behind the forks in RIF1-deficient cells (Fig. 6e, f). These data suggest that the increased ssDNA regions observed at and behind the replication forks in RIF1-deficient cells could be a consequence of defective restart caused due to fork degradation. To verify this hypothesis, we performed a fork restart assay, in which WT and $Rif1^{-/-}$ cells were pre-incubated with DNA2i during HU treatment to prevent fork degradation. Forks were then allowed to restart, and subsequently assessed for IdU tract length (Fig. 6g). DNA2 inhibition during HU treatment completely rescued the restart delay in RIF1-deficient cells (Fig. 6g and Supplementary Fig. 8h). Furthermore, complementation of $Rif1^{-/-}$ MEFs with hRIF1-FL or the Del-HEAT mutant rescued the restart defect in RIF1-deficient cells (Fig. 6h and Supplementary Fig. 8i). However, complementation with either Del-CI or Del-CII did not restore the restart defect upon RIF1 deficiency, in good agreement with our earlier data that these domains are essential for protection of reversed forks (Figs. 6h and 4c, d). These data further strengthen the concept that protection of reversed forks from degradation is linked to efficient fork restart.

Finally, we tested if allowing efficient restart in RIF1-deficient cells could rescue the observed sensitivity to replication stress-inducing agents. $Rif1^{-/-}$ MEFs were complemented with either hRIF1-FL, Del-HEAT, Del-CI or Del-CII, and treated with either HU or cisplatin. Complementation of $Rif1^{-/-}$ with either hRIF1-FL or Del-HEAT significantly rescued the sensitivity of cells to replication stress, in line with our molecular data (Fig. 6i and Supplementary Fig. 6g). In contrast, complementation with Del-CI or Del-CII failed to rescue the sensitivity of RIF1-deficient cells (Fig. 6i and Supplementary Fig. 6g). These results strongly suggest that replication fork protection and subsequent efficient fork restart are physiologically important processes for cellular survival in situations of replication stress.

## Discussion

Our findings identify a novel role of RIF1 in the protection of nascent strands, which underpins how degradation of reversed replication forks can result in genome instability. Replication fork degradation results in genome instability in HR- and FA-defective cells[3,4]. However, it remained poorly understood how degradation of reversed forks results in genome instability.

We show that RIF1 associates with stalled forks and protects them from DNA2-mediated degradation, which is independent of its known interaction with 53BP1[17,19–21,25,26] and thus NHEJ (Figs. 1–3). We further show that loss of $Rif1$ results in de-protection of reversed forks, resulting in extensive fork

degradation (Fig. 2b–f). Importantly, RIF1 was found to act downstream of RAD51, which is involved in fork reversal and in protection of regressed arms (Fig. 2g, h).

Recent studies have also implicated reversed replication forks to be a substrate for MRE11 nuclease action in BRCA2-deficient cells. Other nucleases, including DNA2, MUS81, and EXO1, have also been proposed to mediate fork degradation[7,8,51]. Degradation of reversed forks upon RIF1 deficiency appears to be primarily dependent on DNA2 activity, with a partial requirement of MRE11. These findings suggest that whereas MRE11 can partially access the reversed arm upon RIF1 deficiency, DNA2 is the main nuclease in the degradation process in these conditions (Figs. 2 and 3).

Our data also show that the C-terminal region of RIF1 (consisting of sub-domains CI, CII, and CIII) is essential for protecting reversed forks from degradation (Fig. 4c, d). The CI region has two conserved binding sites for PP1α[40], CII region binds to cruciform DNA structures[43], while the complete C-terminal domain is responsible for BLM binding[48]. Our data demonstrate that the CII domain of RIF1, which binds to cruciform structures, is critical for the protection of reversed forks upon replication stress (Fig. 4c, d). One possibility could be that RIF1 binds to reversed forks, which represent cruciform structures in vivo upon replication stress, and physically protect such forks. Another possibility involves the requirement of both the functions of CI and CII domains of RIF1 in fork protection, as also suggested by our data (Fig. 4c–e). In this scenario, binding of the CII domain to reversed forks could then recruit PP1 through the CI domain to the forks. This recruitment of PP1 could post-translationally restrict DNA2 nuclease activity though de-phosphorylation of DNA2 in the vicinity of forks, thereby protecting them from degradation. In line with this hypothesis, we show that downregulation/ inhibition of PP1 results in reversed fork degradation in WT cells, in a fashion that is epistatic with RIF1 inactivation (Fig. 4e, f). Furthermore, RIF1 inactivation results in hyper-phosphorylation of DNA2 upon replication stress, which again is epistatic with inhibition of PP1 (Fig. 4g). Therefore, one could envision a scenario where access of DNA2 to stalled forks is fine-tuned through PP1-mediated phosphorylation/de-phosphorylation cycles to prevent unrestricted processing of stalled replication forks.

Importantly, our data also provide insight into the mechanisms by which reversed fork degradation results in genome instability. We show that fork degradation upon RIF1 deficiency causes delayed restart, which could be the precursor for subsequent genome instability (Figs. 5 and 6a). RIF1 has multiple roles in the maintenance of genome stability, including in the regulation of origin firing and also in NHEJ. Although disruption of these processes could also contribute to genome instability, our data strongly suggest that delayed restart and subsequent exposure of ssDNA could also contribute to the genome instability upon loss of RIF1 (Figs. 5 and 6).

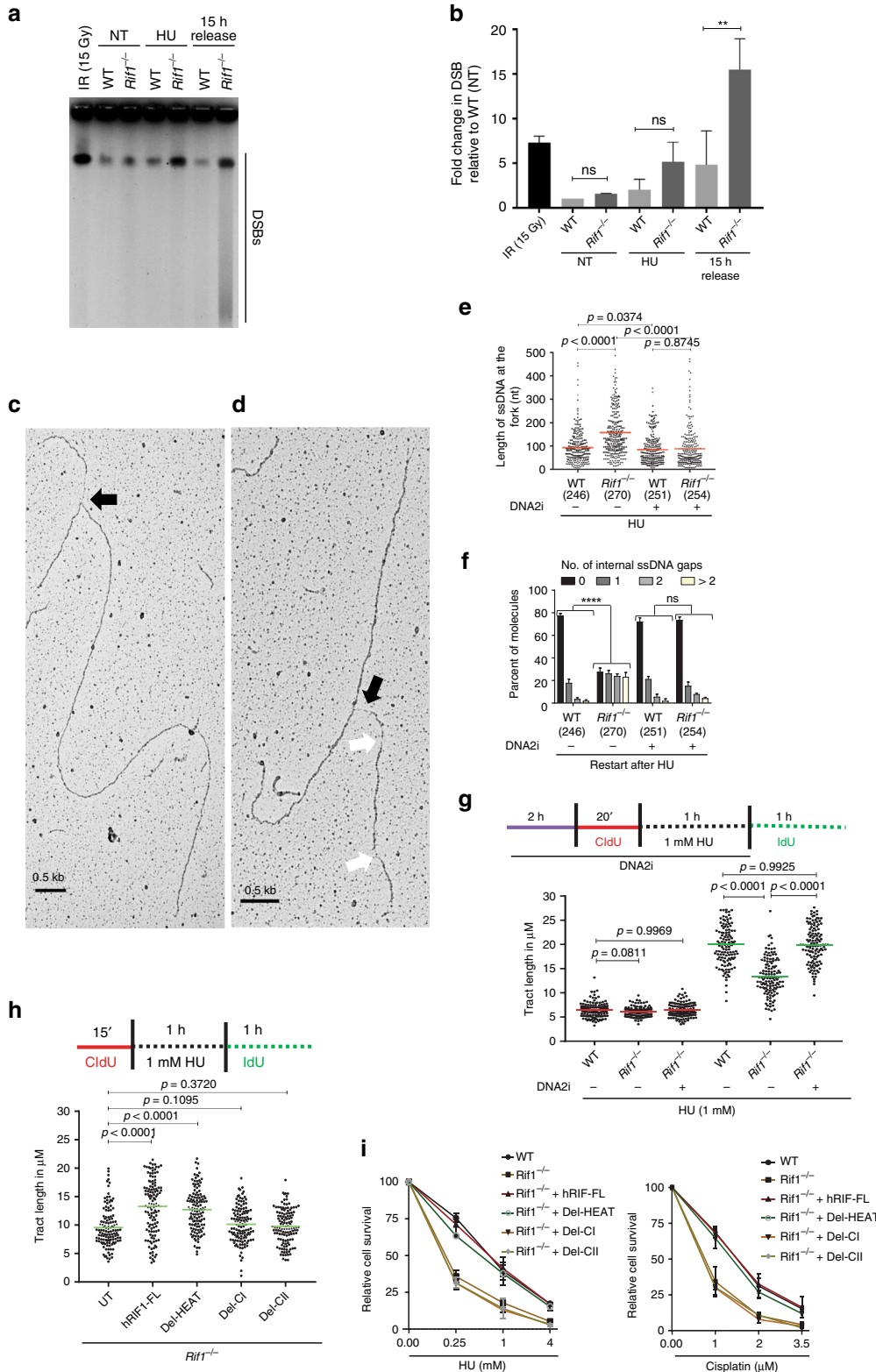

Restart of reversed forks can take place via multiple non-mutually exclusive mechanisms. One mechanism includes helicase-mediated branch migration of the "reversed arm" by RecQ1 helicase[52]. However, upon fork degradation, cells can employ alternate pathways for restart. One such pathway involves re-priming events ahead of the stalled forks. However, re-priming can result in gaps in the daughter strands[46,53]. In line with this speculation, RIF1-deficient cells accumulate increased ssDNA gaps behind the forks when allowed to restart after replication stress. Notably, this phenomenon was dependent on DNA2 activity (Fig. 6c–e). Furthermore, our data indicate that prevention of reversed fork degradation rescues the defective restart in RIF1-deficient cells (Fig. 6f–h). We propose a model, in which RIF1 protects reversed forks from degradation and mediates

**Fig. 6** Restart defects are a consequence of fork degradation in RIF1-deficient cells. **a** PFGE in WT and *Rif1*$^{-/-}$ MEFs with and without treatment with HU for 3 h and 15 h recovery after treatment. **b** Quantification of experiment (**a**), from three independent experiments showing DSB levels relative to WT untreated (NT), (ns, not-significant, **$P = 0.0019$, unpaired *t*-test). **c**, **d** Electron micrographs of ssDNA at the fork (**c**), and behind the fork (**d**), 30 min after release from HU treatment. White arrows represent ssDNA at the forks and black arrows in **d**, represent ssDNA gaps behind the forks **e** Analysis of ssDNA at forks upon restart in WT and *Rif1*$^{-/-}$ MEFs in presence or absence of DNA2 inhibitor. Red bar represents mean, *P*-value was derived from Kruskal–Wallis ANOVA with Benjamini Hochberg post test. **f** Analysis of internal gaps behind forks upon restart in WT and *Rif1*$^{-/-}$ MEFs in the presence or absence of DNA2 inhibitor and HU. Graph represents mean and SD from three independent experiments. Chi-square test of trends was done to assess significance of internal ssDNA gaps between WT and *Rif1*$^{-/-}$ MEFs (ns, non-significant, ****$P < 0.0001$). Numbers of analyzed molecules are indicated within parenthesis for **e**, **f**. **g** Top: schematics for restart assay by fibers upon DNA2 inhibition. Bottom: Tract lengths of IdU and CldU were quantified in WT and *Rif1*$^{-/-}$ MEFs upon restart after treatment with 1 mM HU for 1 h in the presence or absence of DNA2i. **h** Top: schematics for fiber restart assay upon transfection of hRIF1 deletion mutant constructs in *Rif1*$^{-/-}$ MEFs. Bottom: Quantification of IdU tracts in *Rif1*$^{-/-}$ MEFs upon restart after treatment with 1 mM HU for 1 h in the presence or absence of hRIF1 deletion constructs. Red and green bars in **g** and **h** represents mean CldU and IdU tract length, *P*-values were obtained from Kruskal–Wallis ANOVA with Benjamini Hochberg post test for FDR. All experiments were repeated thrice (Supplementary Data 2 and Supplementary Fig. 8h–i). **i** Survival assay in *Rif1*$^{-/-}$ MEFs complemented with hRIF1-FL, Del-HEAT, Del-CI, Del-CII constructs of hRIF1 and treated with different concentrations of HU and Cisplatin. Data represented from three independent experiments, error bars represent s.e.m

efficient restart due to the presence of the reversed arm as substrate for branch migration (Fig. 7a, b). Absence of RIF1 leads to extensive fork degradation, resulting in delayed restart. This delayed fork restart results in the exposure of under-replicated DNA behind the forks (Fig. 7c). The under-replicated DNA then becomes a source of genome instability later (Fig. 7c).

Identification of the mechanisms underlying replication fork degradation is also clinically relevant, as fork protection in BRCA-deficient tumors has recently been implicated in chemoresistance[5,6]. We speculate that fork degradation at difficult-to-replicate regions of the genome could be a potential source of genome instability. Consistent with this idea, RIF1-deficient cells show a slightly higher background level of genome instability (Fig. 5a). These low -but tolerable- levels of genome instability combined with a checkpoint defect could result in accelerated tumorigenesis. On the other hand, cancers with *RIF1* mutations could be more responsive to chemotherapeutic regimens. Although further studies are required to test these hypotheses, mechanistic insights into the process of replication fork protection could result in the development of potentially new therapeutic regimens for cancer.

## Methods

**Cell culture, cell lines, and transfection reagents**. All the MEFs (WT, *Rif1*$^{-/-}$, *53bp1*$^{-/-}$, and 53BP1[15A])[30] were cultured in Dulbecco's Modified Eagle Medium (DMEM) supplemented with 10% fetal calf serum (FCS) and 1% penicillin–streptomycin (PS, P0728 Sigma) at 37 °C and 5% in a humidified incubator. Transfections were performed using transfection reagents Xtremegene-9 (Roche) and Lipofectamine-2000 according to the manufacturer's protocol. WT and RIF1-KO HAP1[27] were cultured in Iscove's Modified Dulbecco's Media (IMDM) containing 10% FCS and 1% pen–strep.

**Generation of deletion mutants**. RIF1 mutants were created using the standard PCR and cloning methods. The following primers were used for creating the deletion mutants for various domains of human RIF1:

hRif-DelCTD-Rev : 5′-GACACAGCGTGTCTGCA-3′
hRif-DelCTD-Fwd : 5′-TAGGACCCAGCTTTCTTGTAC-3′
hRif-DelHEAT- Rev : 5′- CATGGTGAAGCCTGCT-3′
hRif-DelHEAT-Fwd : 5′- CCTGGTTTGGAAACTGTTGAAAT-3′
hRif-DelC1-Fwd : 5′-CAATCTAAGATTTCAGAAATGGCCA-3′
hRif-DelC2-Rev : 5′- GTTCACCAATGGTGGGTAAACA -3′
hRif-DelC2-Fwd : 5′- CTAGAAGAGATTCCAGTTTTTGATATTTCT -3′

The GFP-RIF1 constructs used in this study is based on pcDNA5/FRT/TO-GFP-RIF1 described previously[24], which has human RIF1 cDNA fused to GFP at its N-terminus. Domain deletions were created using Q5 Mutagenesis Kit (NEB, cat. No# E0554S), following the manufacturer's instruction. Primers were used to PCR amplify the entire plasmid leaving out the region of RIF1 to be deleted. PCR products were gel purified and ligated. Introduction of domain deletions were further verified by Sanger sequencing.

**iPOND-SILAC mass-spectrometry**. For SILAC labeling, mouse embryonic cells (mESCs) were maintained in serum free 2i media deficient in lysine, arginine, and

L-glutamine (PAA) at 37 °C and 5% CO2 in a humidified incubator. Cells were grown in medium containing either 73 μg/ml light [$^{12}C_6$]-lysine and 42 μg/ml [$^{12}C_6$, $^{14}N_4$]-arginine (Sigma) or similar concentrations of heavy [$^{13}C_6$]-lysine or [$^{13}C_6$, $^{15}N_2$]-lysine and or [$^{13}C_6$, $^{15}N_4$]-lysine arginine (Cambridge Isotope Laboratories).

For iPOND experiments, cells were labeled with 10 μM EdU for 10 min and then treated with HU (4 mM) for 2 h to stall the DNA replication forks. After labeling and treatment cells were washed with Phosphate Buffer Saline (PBS) and harvested using cell scrapper. Samples were then treated with click reaction containing 25 μM biotin-azide, 10 mM ( + ) sodium L-ascorbate and 2 mM CuSO$_4$ and rotated at 4 °C for 1 h. Samples were then centrifuged to pellet down the cells; supernatant was removed and replaced with 1 ml Buffer-1 containing 25 mM NaCl, 2 mM EDTA, 50 mM Tris–HCl, pH 8.0, 1% IGEPAL and protease inhibitor and rotated again at 4 °C for 30 min This step was repeated twice. Samples were centrifuged to pellet down the cells; supernatant was removed and replaced with 500 μl of B1 and sonicated 30 times for 20 s on and 90 s off at high amplitudes using a Diagenode Bioruptor plus sonicator. Samples were centrifuged, and supernatant was transferred to fresh tubes and incubated for 1 h with 200 μl of Dyna-Beads My-One C1 for the streptavidin biotin capture step. Proteins were eluted, and mass-spectrometry was performed. At least two peptides were required for protein identification. Quantitation is reported as the log$_2$ of the normalized heavy/light ratios. SILAC data were analyzed using MaxQuant. The resulting output tables of two independent experiment were merged and used as the input for calculating the average fold-change to identify significantly upregulated proteins upon HU treatment based on H:L ratio in the SILAC experiment in the MaxQuant software[54].

**Immunoblotting**. Cells were lysed in 4x Laemmli sample buffer and boiled for 5 min. Proteins were separated on 4–12% NuPAGE Bis-Tris Gel (Novex life technologies) and transferred on nitrocellulose membrane (0.45 μM). Membranes were blocked with 5% milk in PBS-1% Tween20 for 1 h and incubated overnight in primary antibodies. Membranes were then washed three times with PBS containing 0.05% tween and probed with respective secondary antibodies. Finally ECL Prime Western Blotting Detection Reagent kit (GE Healthcare) was used to develop the blots. Details of the antibodies used are provided in Supplementary Table 1.

**DNA fiber analysis**. DNA fiber analysis was carried out according to the standard protocol as mentioned previously[18]. Briefly, cells were sequentially pulse-labeled with 30 μM CldU (c6891, Sigma-Aldrich) and 250 μM IdU (I0050000, European Pharmacopoeia) for 20 min and treated with HU (4 mM) for 3 h for fork degradation assay, and for fork restart assay after first labeling with CldU cells were treated with 1 mM HU for 1 h. After labeling, cells were collected and resuspended in PBS at $2.5 \times 10^5$ cells per ml. The labeled cells were mixed at 1:1 (v/v) with unlabeled cells, and 2.5 μl of cells were added to 7.5 μl of lysis buffer (200 mM Tris-HCl, pH 7.5, 50 mM EDTA, and 0.5% (w/v) SDS) on a glass slide. After 8 min, the slides were tilted at 15–45°, and the resulting DNA spreads were air dried, fixed in 3:1 methanol/acetic acid overnight at 4 °C. The fibers were denatured with 2.5 M HCl for 1 h, washed with PBS and blocked with 0.2% Tween 20 in 1% BSA/PBS for 40 min The newly replicated CldU and IdU tracks were labeled (for 2.5 h in the dark, at room temperature (RT)) with anti-BrdU antibodies recognizing CldU (1:500, ab6326; Abcam) and IdU (1:100, B44, 347580; BD), followed by 1 h incubation with secondary antibodies at RT in the dark: anti–mouse Alexa Fluor 488 (1:300, A11001, Invitrogen) and anti-rat Cy3 (1:150, 712-166-153, Jackson Immuno-Research Laboratories, Inc.). Fibers were visualized and imaged by Carl Zeiss Axio Imager D2 microscope using 63X Plan Apo 1.4 NA oil

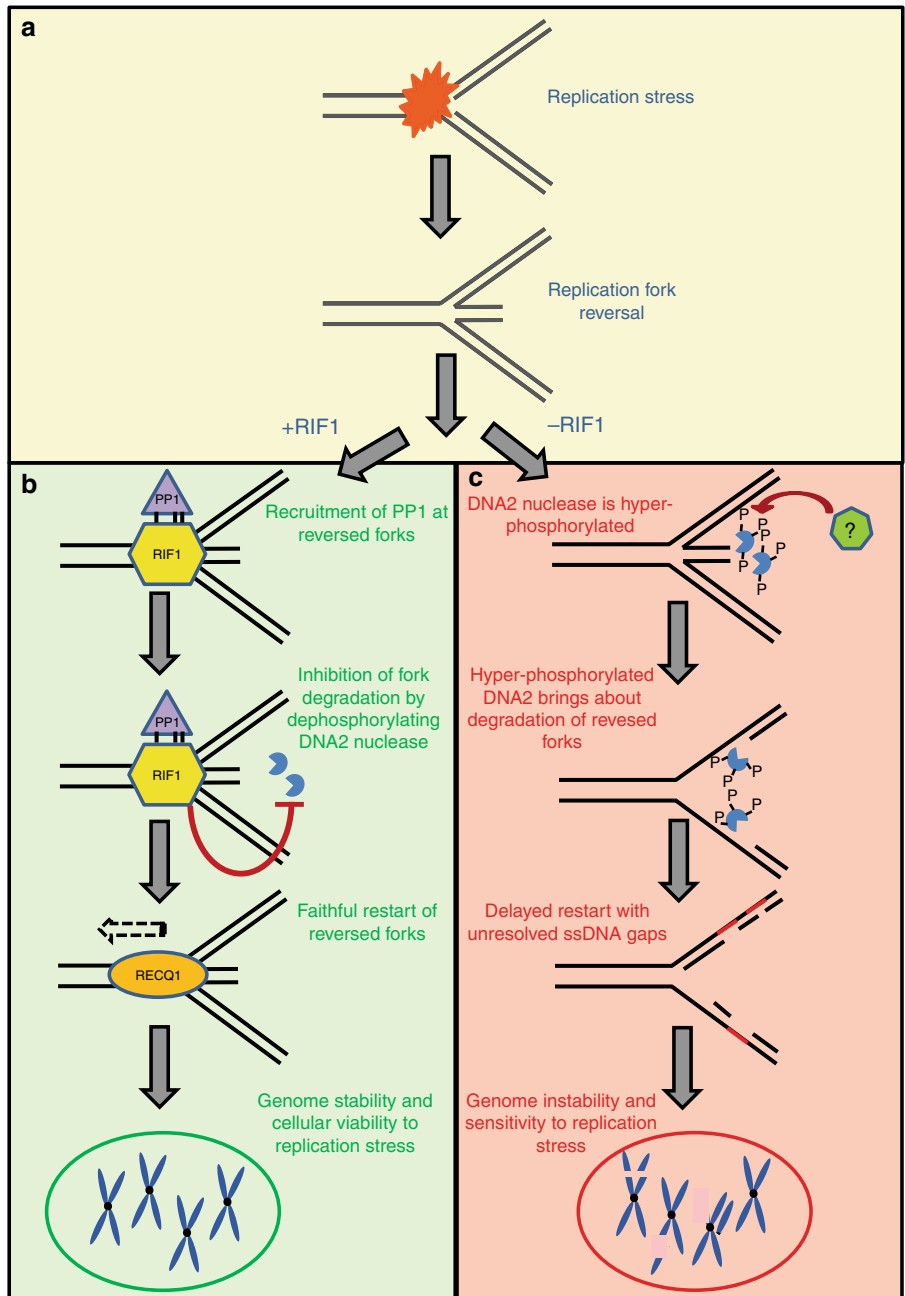

**Fig. 7** Model for role of RIF1 in fork protection and genome stability. **a** Replication stress in cells results in replication fork reversal to stabilize stalled replication forks. **b** Fork reversal results in the recruitment of RIF1 probably through its C-terminal domain, which has cruciform structure binding properties. Binding of RIF1 to reversed forks stabilizes them by recruitment of PP1, which brings about de-phosphorylation of DNA2 and thereby limits access of DNA2 nuclease to these forks and prevents fork degradation. This allows for normal restart of reversed forks probably through RECQ1-mediated branch migration of these reversed forks resulting in prevention of genome instability and cellular viability upon replication stress. **c** In contrast, absence of RIF1 results in DNA2-mediated degradation of reversed forks. In the absence of the preferred substrate (four-way junctions), RECQ1 is unable to bind. Forks are therefore aberrantly restarted which results in exposure of under-replicated DNA in the form of ssDNA gaps behind the forks. These ssDNA gaps become a source of genome instability and DSBs later during the cell cycle in G2/M phases resulting in sensitivity to replication stress-inducing agents

immersion objective. Data analysis was carried out with ImageJ software[64]. The Mann–Whitney test was applied for statistical analysis using the GraphPad Prism Software.

**Colony survival assay**. Colony survival assay was performed according to the standard protocol as previously mentioned[55]. WT and Rif1$^{-/-}$ MEFs were seeded at low dilutions and treated with different replication poisons (HU, CPT, and Cisplatin) with different concentrations for 4 h. In complementation experiments, first Rif1$^{-/-}$ MEFs were transfected with hRIF deletion constructs along with full-length (hRIF1-FL, Del-HEAT, Del-CI, and Del-CII). The protein expression was allowed for 48 h and confirmed by western blotting. In parallel same cells were plated out at low dilutions and treated with drugs at different concentrations for 4 h. Post treatment, drug treated medium was washed out and cells were allowed to grow in complete growth medium for 8 days. The colonies detected were fixed, stained, and subsequently analyzed with the Gel-counter by Oxford Optronix and appertaining Software (version 1.1.2.0). The survival was plotted after combining three independent experiments as the mean surviving percentage of colonies after drug treatment compared to the mean surviving colonies from the non-treated samples.

**Metaphase spreads and chromosomal aberrations**. Metaphase spreads were carried out according to the standard protocol described previously[8]. Briefly, exponentially growing cells (50–80 % confluence) were treated with drugs at different concentrations for 4 h. Post treatment, drug treated medium was washed out and cells were allowed to grow in complete growth medium and exposed with colcemid for 6 h. Metaphase spreads were prepared, stained by conventional methods. A minimum 60 metaphase images were using Carl Zeiss Axio Imager D2 microscope using 63x Plan Apo 1.4 NA oil immersion objective and analyzed with ImageJ software64 for chromosomal aberrations. Experiments were repeated three times. To determine the differences between conditions are significant a two-tailed $t$-test was used. $P$-values $< 0.005$ (***) and $< 0.0005$ (****) were considered as significant.

**Immunofluorescence**. Cells were grown on coverslips at 70–80% confluency. Cells were labeled with EdU (10 μM) for 15 min to visualize cells in S-phase. After 15 min, cells were washed and incubated in fresh medium containing HU (4 mM) for 3 h. After 3 h cells were washed in PBS and fixed in 2% paraformaldehyde in PBS containing 0.1% Triton-X and permeabilized for 15 min in 0.5% Triton-X in PBS. Coverslips were washed three times and subsequently stained with RIF1 primary antibody (1:5000, rabbit) for 1 h at RT. After 1 h, cells were washed with PBS containing 0.1% Triton-X followed by incubation in secondary antibody conjugated to Alexafluor-488 for 1 h at RT. EdU was visualized with a click-it reaction (click-it EdU imaging kit, Invitrogen) using a 594 nM fluorescent azide according to the manufacturer's protocol. Coverslips were washed and incubated with DAPI (4',6-diamidino-2-phenylindole) for 10 min and mounted with ProLong Gold antifade reagent (Invitrogen). In experiments using ionizing radiations, cells were irradiated with (10 Gy) and allowed to recover for 2 h followed by 2% paraformaldehyde fixation and immunostaining. Images were obtained using Carl Zeiss Axio Imager D2 microscope using 63X Plan Apo 1.4 NA oil immersion objective and analyzed with ImageJ software64. Details of the antibodies are provided in Supplementary Tables 1 and 2.

**Electron microscope analysis**. EM analysis was performed according to the standard protocol[38]. For DNA extraction, cells were lysed in lysis buffer and digested at 50 °C in the presence of Proteinase-K for 2 h. The DNA was purified using chloroform/isoamyl alcohol and precipitated in isopropanol and given 70% ethanol wash and resuspended in elution buffer. Isolated genomic DNA was digested with PvuII HF restriction enzyme for 4 to 5 h. Replication intermediates were enriched by using QIAGEN G-100 columns (as manufacture's protocol) and concentrated by an Amicon size-exclusion column. The benzyldimethylalkylammonium chloride (BAC) method was used to spread the DNA on the water surface and then loaded on carbon-coated nickel grids and finally DNA was coated with platinum using high-vacuum evaporator MED 010 (Bal Tec). Microscopy was performed with a transmission electron microscope FEI Talos, with 4 K by 4 K cmos camera. For each experimental condition, at least 70 replication fork intermediates were analyzed per experiment and ImageJ software64 was used to process analyze the images.

**DSB detection by PFGE**. DSB detection by PFGE was done as reported previously[18]. Briefly, cells were cast into 0.8% agarose plug ($2.5 \times 10^5$ cells/plug), digested in lysis buffer (100 mM EDTA, 1% sodium lauryl sarcosine, 0.2% sodium deoxycholate, 1 mg/ml proteinase-K) at 37 °C for 36–40 h, and washed in 10 mM Tris-HCl (pH 8.0)–100 mM EDTA. Electrophoresis was performed at 14 °C in 0.9% pulse field-certified agarose (Bio-Rad) using Tris-borate-EDTA buffer in a Bio-Rad Chef DR III apparatus (9 h, 120°, 5.5 V/cm, and 30- to 18-s switch time; 6 h, 117°, 4.5 V/cm, and 18- to 9-s switch time; and 6 h, 112°, 4 V/cm, and 9- to 5-s switch time). The gel was stained with ethidium bromide and imaged on Uvidoc-HD2 Imager. Quantification of DSB was carried out using ImageJ software64. Relative DSB levels were calculated by comparing the results in the treatment conditions to that of the DSB level observed in untreated controls.

**Flow cytometry for cell cycle analysis and detection of RPA**. For flow-cytometric analysis of cell cycle, cells were labeled with EdU for 20 min followed by fixation for 10 min in 4% formaldehyde/PBS at room temperature. Cells were then washed with 1% BSA/PBS and permeabilized in 0.5% saponin buffer in 1% BSA/PBS. Incorporated EdU was labeled according to the manufacturer's instructions (#C35002; Invitrogen). Detection of RPA levels were carried out as previously described[56]. Briefly, cells were harvested after treatment with 4 mM HU in 1 h and the mentioned release time points. Pre-extraction was carried out in 0.2% Triton-X-100/PBS to remove the non-chromatin bound RPA. Cells were then washed with 0.1/BSA/PBS and fixed with 4% Paraformaldehyde. Permeabilization was carried out as discussed above, followed by staining with anti RPA32/2 antibody (#ab2175, Abcam) for 1.5 h and subsequent incubation in secondary antibody. In both the assays, DNA was stained with 1 μg/ml DAPI. Samples were measured in BD LSR Fortessa and analyzed by FlowJo software v10.5.0. Details of the antibodies are provided in Supplementary Tables 1 and 2.

**Proximity ligation-based assays (PLA)**. Cells were grown on cover slips until 60–70% confluency. Cells were incubated with EdU (20 μM) for 15 min to visualize cells in S-phase. After 15 min cells were washed and incubated in fresh media containing HU (4 mM) for 3 h. After 3 h cells were washed two times with PBS and fixed with 2% paraformaldehyde for 15 min at room temperature. Following fixation and washing, cells were next permeabilized by 0.5% Triton-X in PBS for 15 min at room temperature. After two washes, freshly prepared click reaction mix (2 mM copper sulfate, 10 μM biotin-azide, and 100 mM sodium ascorbate in PBS) was added to each samples and incubated in a humidified chamber at room temperature for 1 h. After the click reaction, slides were washed with PBS for 5 min and placed back in humidified chamber with blocking buffer (10% goat serum and 0.1% Triton-X-100 in PBS) for 1 h at room temperature. Slides were incubated with primary antibody at 4 °C overnight in a humidified chamber. Mouse anti-biotin was used in conjunction with the RIF1 antibody. Slides were washed three times with wash buffer A (0.01 M Tris, 0.15 M NaCl, and 0.05% Tween 20, pH 7.4) for 5 min each. Duolink In Situ PLA probes anti–mouse plus and anti–rabbit minus were diluted 1:5 in blocking solution (10% goat serum and 0.1% Triton-X-100 in PBS), dispensed onto slides (30 μl/well), and incubated for 1 h at 37 °C. Slides were again washed three times with buffer A 5 min each. Ligation mix was prepared by diluting Duolink ligation stock (1:5) and ligase (1:40) in high-purity water. Slides were placed back in the humid chamber, and ligation mix was dispensed onto slides (30 μl/well) and incubated at 37 °C for 30 min Slides were washed in 60 ml wash buffer A two times for 2 min each. Amplification mix was prepared by diluting Duolink amplification stock (1:5) and rolling circle polymerase (1:80) in high-purity water. Slides were placed back in the humid chamber, and amplification mix was dispensed onto slides (30 μl/well) and incubated at 37 °C for 100 min. Slides were washed with wash buffer B solution (0.2 M Tris and 0.1 M NaCl) three times for 10 min each and one time in $0.01 \times$ diluted wash buffer B solution for 1 min, coverslips were incubated with DAPI for 5 min and mounted with ProLong Gold antifade reagent (Invitrogen). Slides were imaged using Zeiss LSM 700 Axio Imager Z2 confocal microscope and analyzed using ImageJ software64.

**Immunoprecipitations**. Cytoplasmic and nuclear extracts were prepared using NE-PER Nuclear and Cytoplasmic extraction kit according to manufacturer's protocol. Protein concentrations were determined using BCA kit. Protein samples were precleared using 20 μl of Protein A Sepharose Fast Flow (PAS) beads (GE Healthcare) by incubating 30 min at 4 °C. Samples were spun down and supernatant was collected in fresh tube. Ten percent inputs were taken out and 200 μg of nuclear proteins were incubated with 2 μg of DNA2 antibody along with respective IgG control over night 4 °C keeping final volume of 500 μl with RIPA buffer (0.01 M Tris-HCl pH 7.5, 0.15 M NaCl, 1% Triton-X-100, 1% NP-40, 0.1% SDS, protease inhibitor cocktail). Next day, 50 μl of Protein A Sepharose washed beads with 1XPBS + 0.1% NP40 was added and samples were further incubated for 2 h at 4 °C. After final incubation samples were washed five times with wash buffer (150 mM NaCl, 20 mM Tris-HCl (pH 7.5), 0.1% NP-40, 1 mM EDTA, 2.5 mM Sodium pyrophosphate, protease inhibitor) 500 μl each, and eluted in 2x Laemmli sample buffer for sodium dodecyl sulfate polyacrylamide gel electrophoresis and immunoblot.

**Plasmid-based DNA repair assays**. For determining the HR and NHEJ proficiency in WT and $Rif1^{-/-}$ cells the following plasmids were used: pDRGFP (Addgene plasmid #26475) and pCBASceI (Addgene plasmid #26477) for HR and pCVL Traffic Light Reporter1.1(Sce target) (Addgene plasmid #31482) along with pCBASceI for NHEJ. For measuring and controlling transfection efficiency, Turbo-GFP expressing plasmid (Sigma, MISSION SHC003) and Scrambled plasmid (Sigma, MISSION SHC002), were used. In all, $2.5 \times 10^5$ cells were co-transfected with 3 μg of plasmid combinations (i.e., 1.5 μg of each plasmid) using Xtremegene-9 reagent from Roche in six-well dish. Cells were transfected twice at an interval of 24 h. Post 48 h of transfection, cells were harvested and the GFP-positive cells (for HR) and RFP-positive cells (for NHEJ) were assessed by flow cytometry. Fifty thousand events were recorded for each sample. Background normalization was done by using samples co-transfected with scrambled—reporter plasmid and also scrambled—pCBASceI plasmid. Final percentage of GFP and RFP-positive cells were calculated based on the transfection efficiency of the cells. Each experiment was independently performed at least thrice.

**Reporting summary**. Further information on research design is available in the Nature Research Reporting Summary linked to this article.

## Data availability

The iPOND dataset has been deposited in Figshare repository [https://doi.org/10.6084/m9.figshare.8242088.v1]. The source data underlying Figs. 3a, 4b, g, 5a, 6a, and Supplementary Figs. 1b, 2a, b, e, 3c, f, g, 4c, and 6a, c are provided as a Source Data file. A reporting summary for this Article is included as Supplementary Information file.

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

## Acknowledgements

We thank Roland Kanaar, Andre Nussenzweig, Massimo Lopes, and Nitika Taneja for stimulating discussions and sharing important reagents used in the manuscript; Alessandro Vindigni and Grzegorz Ira for DNA2i, Michela Di Virgilio for mouse RIF1 antibody, Calvin Lo for help with flow cytometry, Przemek Krawczyk for constructs, Ross Chapman and Simon Boulton for cell lines. This work was supported by grants from the Netherlands Organization for Scientific research (NWO-VIDI #91713334) and the

European Research Council (ERC CoS Grant 682421) to MATMvV, a Dutch Cancer Society (KWF grant 11008/2017-1) grant, Erasmus MC Daniel den Hoed instrument grant and startup funds from the Erasmus MC to ARC.

## Author contributions

C.M. conducted all the fiber experiments, PFGE, FACS, and cloning experiments. V.T. performed clonogenic assays, metaphase spreads, PLA and immunofluorescence experiments. E.M.M. performed the EM experiments with help from C.M. A.M.H. and H.R.d.B. performed SCE experiments. G.R. performed immunofluorescence experiments for RAD51. S.M. helped C.M. in cloning experiments. J.D. analyzed mass-spectrometry data. M.A.T.M.v.V supervised the SCE experiments and helped in finalizing the manuscript. A.R.C. conceptualized the project, supervised it, and wrote the manuscript.

## Additional information

**Competing interests:** The authors declare no competing interests.

