## [Peer Review File · Nature Communications]

Reviewers' comments:

Reviewer #1 (Remarks to the Author):

The authors in this paper make the observation that the Rif1 protein is needed to protect stalled replication forks from undergoing extensive, DNA2-dependent, degradation. This correlates with a small defect in fork restart and hypersensitivity to hydroxyurea and cisplatin. The authors also find that the Rif1 interaction with PP1 but not 53BP1 may be important. These observations add Rif1 to the very long (and growing) list of proteins that are needed to prevent fork degradation.

What is not answered here is how Rif1 protects replication forks from nuclease action. The authors suggest it is through PP1, which presumably has many substrates that could be relevant. Fully answering this question may be very difficult. But some simple experiments could be done to try to understand the pathway.

There are a large number of experimental and textual issues that need to be addressed:

1. Is the function of Rif1 in fork protection upstream of stabilizing a RAD51 filament on the reversed forks? This is an important question to answer since some but not all fork protection pathways operate upstream of RAD51.
2. The iPOND data sets described on page 5 must be included in the manuscript. The heavy/light ratios from each experiment should be reported for all proteins observed in the mass spectrometry experiment. This information is needed to evaluate the claims based on this data.
3. The Bunting lab found 53BP1 to be localized to stalled replication forks by iPOND (Her et al., MCB 2018). Is 53BP1 found in this iPOND dataset? Does it co-localize with Rif1 after HU to EdU foci?
4. The Bunting lab also showed that 53BP1 deficient cells have nascent strand degradation. This is in contrast to what is presented in figure 2d. How many 53bp1^{-/-} cell clones were analyzed in this figure? Is it possible that the cells used picked up a suppressor mutation? Please provide evidence that the 53bp1^{-/-} cells are actually 53bp1-deficient. This is important for the localization studies in Figure 1 as well as the model of fork protection.
5. The Her paper (Her et al., MCB 2018) should be referenced or discussed to put the 53BP1 differences in this manuscript in context with what was previously published.
6. Buonomo et al., JCB 2009 showed that Rif1 deficient cells were hypersensitive to replication stress (aphidicolin and MMC). Thus, it is not surprising to find they are also sensitive to HU. This finding should be referenced.
7. Buonomo et al., also showed that Rif1-deficient cells are defective in HDR. This contrasts with this paper that found no defect in RAD51 foci. The authors should do a more direct measurement of HR repair of DSBs using one of the well-established reporter assays to try to reconcile the different results.
8. Xu et al., EMBO 2010 reported that Rif1 deficient cells were sensitive to HU. This paper should be referenced.
9. Xu et al., ELIFE 2017 further showed that Rif1 and 53BP1 act in the same pathway to maintain viability in HU treated cells. This paper needs to be referenced and discussed since it contrasts with the authors model.
10. The EM data in figure 2f is not convincing. First, the percentage of reversed forks in WT cells is

much higher than previously reported by other investigators. Second it is not clear if there is any difference between the HU-treated WT and Rif1^{-/-} cells since there is no measurement of variability. The authors state that similar results were observed in “at least one independent experiment”. Please present those data. Similar concerns need to be addressed for figure 3d.

11. The authors should show the effect of PP1i treatment on fork degradation in Rif1^{-/-} cells. The prediction from their model is that PP1 and Rif1 should be epistatic.

12. The authors suggest that CDK1 phosphorylation of DNA2 is important for it to degrade forks. This is based on a single experiment (Figure 4f). Much more would need to be done to validate this idea. Does CDK1i treatment affect DNA2 localization to stalled forks? Does it affect fork reversal? What happens when analyzing a DN2 mutant that cannot be phosphorylated? Does it affect fork degradation in BRCA2^{-/-} cells? Etc.... This should either be removed or further experiments should be done to understand what CDK1 is doing in this pathway. After all, there may be many CDK1 targets that are important.

13. The results in Figure 6e (top panel) with the DNA2i look to me like it does not rescue the ssDNA region formation effectively. This is in contrast to the complete restoration of fork protection. My interpretation is the opposite of the authors – this data suggests that the fork degradation and generation of ssDNA regions are not linked.

14. The conclusion that protection of reversed forks from degradation “is tightly coupled” to fork restart is an over-interpretation of the data. First, there is only a slight fork restart defect observed and that defect is in the speed of elongation. Second, other fork protection defects (ie BRCA2) do not show any defects in restart. The differences with BRCA2 should be discussed with reference to the Schlacher/Jasin paper originally describing fork protection.

15. The conclusion that fork restart is defective uses a 1hr recovery time. In these conditions, the authors see a small reduction in length of the label added during the 1h treatment. So restart is actually fine but the authors conclude the restarted forks are slow. However, if inter-origin distance was shorter in the Rif1 mutant cells then it is possible the shorter second label is just because the forks didn't need to move as far to complete replication. A shorter time course would help alleviate this concern.

16. A larger point is that the authors make many interesting observations that correlate with each other (ie sensitivity to damage/HU and fork degradation) which could suggest a relationship but they overstate this conclusion. It is also possible that these biological effects of Rif1 inactivation are independent of each other. Considering the large numbers of things that Rif1 has been suggested to do and the pleiotropic functions of PP1, this wouldn't be unexpected. The text and conclusions need to better reflect these uncertainties.

17. The model and discussion includes the idea that the problem in Rif1 defective cells is due to “aberrant restart through repriming events”. There is no evidence presented in the paper or any experiments completed that addressed “repriming”. The authors do measure ssDNA gaps by EM the origin of those gaps is unclear. I think speculation on this point is fine in the discussion but it should be removed from the model figure and the text needs to either mention other possibilities or make clear that this is speculation.

18. The statistical analyses need to be done with more appropriate tests. A Mann-Whitney test is not appropriate when comparing more than 2 samples. A test that takes into account false-discovery rates is needed. This applies to nearly all of the fork protection assays presented since more than 2 samples are compared.

19. I don't understand the point in the discussion about nuclease polarity. Presumably the authors

assays are monitoring degradation of both the leading and lagging nascent DNA strands so how could a difference in nuclease polarity be important?

20. How long were the cells treated with HU in Figure 1d? Was it less than what would induce DSBs?

21. Page 5: "consistent with earlier published data..." A reference is needed here.

22. The iPOND data was analyzed with a "two-sided T test". However, the authors only completed two biological replicates. I don't think n=2 would provide sufficient statistical power to make a t-test useful.

23. Page 4 "mechanisms...remains"

24. Page 4: "MCM2 deficient mice which are viable..." This is incorrect. The MCM2 mice are not null alleles.

Reviewer #2 (Remarks to the Author):

Review comments:

This manuscript by Mukherjee et al. describes RIF1 being enriched at stalled replication forks and inhibits DNA2 nuclease activity towards reversed replication forks. This protection of replication forks was dependent on RIF1 interaction with Protein Phosphatase1 and cruciform structures but not RIF1 function in NHEJ. RIF1 deficiency delayed fork restart and increased ssDNA formation and reduced survival in response to replicative stress. Thus, RIF1 play an important role in replication fork protection upon replication stress and therefore contribute to genome stability. Most of the experiments are well designed with proper controls. The data is convincing. Replication stress is believed to be critical for cancer initiation, therefore, stabilization of DNA replication forks is also believed to be essential for maintenance of genome stability and prevention of tumorigenesis. The most widely studied role of RIF1 is in the repair of DSBs via NHEJ, and via its interaction with 53BP1. The hypothesis that a 53BP1 independent function of RIF1 at stalled replication forks, inhibition of DNA2 nuclease activity towards reversed replication forks, provide an unexpected but biologically reasonable potential explanation. Therefore, the proposed research is both biologically and clinically important and interesting.

However, the manuscript by Mukherjee et al. is not written carefully. Multiple figures lack error bars and statistical analysis. The number of repeats and how many fiber/ssDNA/reversed fork/abnormal chromosome were quantified is not mentioned in some cases. In addition, the overall evidence supports most of their claims, but still the authors tend to overinterpret the data. For example, the role of PP1 or CDK1 is really tangential to the story presented, and the connection to RIF1 is unclear.

Here are some specific points:

The major concern:

1. Figure 1d, there is no evidence that these newly synthesized DNA foci are stalled replication forks. Evidence for RIF1 positive foci at stalled replication forks is needed.

2. Figure 3. Only data suggests involvement of DNA2 as DNA2 inhibitor rescues fork degradation in RIF1 deficient cell. DNA2 is known to drives processing and restart of reversed replication forks in human cells, DNA2 inhibitor can rescue fork degradation in RIF1 deficient cell in many possible ways that is indirect. Evidence needed to demonstrate RIF1 directly interacts with DNA2 in vivo

and invitro and inhibit DNA2 nuclease activity in vivo or in vitro.

3. Figure 4. The impact of Both PP1 inhibition and CDK1 inhibition are broad and inconclusive. Figure 4e and 4f only provide correlative supporting evidence. CDK1 phosphorylation site on DNA2 is known, phospho-mimetic mutant and phospho-null mutant of DNA2 can provide more direct and specific convincing evidence in both % of reversed forks and fork degradation DNA fiber assay to support conclusion of figure 4e and figure 4f.

4. Figure 6a, Increased DSB in RIF1-/- can be due to defect in DSB repair during recovering from HU induced replication stress. Evidence needed to demonstrate the increased DSB is not all due to decreased DSB repair. Supplementary figure 3a 3b only demonstrate change in HR is not significantly affected, evidence showing that NHEJ is not affected is needed.

5. Supplemental Figure 6, interpret RPA staining as marker for ssDNA in flow cytometry experiment is difficult because only chromatin bound RPA can be used to evaluate the amount of ssDNA. The RPA result also contradict the EM results. EM result in figure 6e shows huge increase in ssDNA 30 minutes after release from HU, while RPA results shows similar ssDNA 5 hours after release and increase after 15 hours release.

Minor concerns:

1. Provide table with list of top hits with p-value FDR etc. for figure 1b.
2. Figure 2f, no error bar not statistics analysis. no number of repeats
3. Figure 3d, no error bar not statistics analysis. no number of repeats
4. Figure 4d, no error bar not statistics analysis. no number of repeats
5. Figure 6a, quantification and represented image are very contradictory. The image shows IR have much higher DSB than rif1-/- 15hr release but quantification suggest otherwise. Also, WT HU have higher DSB than WT 15 hrs release, but quantification suggest otherwise. Quality of representative image is low and longer exposure may better demonstrate the result.
6. Figure 6e, no error bar, no statistics analysis, no number of repeats or number of molecules were quantified.
7. The quality of WB in supplementary figure 2a and 2c is poor and needs to be repeated.

Reviewer #3 (Remarks to the Author):

The manuscript by Mukherjee and colleges describes a role for RIF2 in preventing degradation of reversed replication fork that involves its ability to bind PP1 but is independent of 53BP1. Rif1 specifically prevents DNA2 from attacking reversed forks to allow efficient restart. This is an interesting new role for RIF1 that should be of interest to the genome maintenance and replication fields. Overall, the authors present convincing evidence to support their conclusions and, given that the critical points below concerning data availability and statistics are addressed, I recommend this work for publication.

Concerns

1. I could not find the results of the mass spec experiments in the manuscript. This is an important resource from this work and must be included to allow comparison of their iPOND data to other published data sets and validate the quality of the experiments. The full list of proteins with the SILAC ratios should be provides as a table and the raw ms data should be deposited in a proteome

data repository according to best practice in the field.

2. In many experiments it is unclear whether statistics includes biological variation between independent experiments or merely variation between observations in one experiment. This should be clarified, and as a minimum the authors should confirm that two biological independent experiments showed similar results for:

Fig. 2a, b, c, d

Fig. 3b, c

Fig. 4 c, e, f

Fig. 5h

Fig. 6 f,g

For the EM experiments, statistics are missing for the following experiments and it is unclear how many biological replicates they represent

Fig. 2e

Fig. 3d

Fig. 4d

Fig. 6e

3. In the EdU facs analysis supplementary fig. 2a there appears to be more S phase cells with low EdU incorporation in the absence of RIF1 and the distribution of cells within S phase also appears to be skewed towards early S compared to wt cells. This might reflect changes in replication timing or that a higher fraction of cells experiences problems during normal replication. The authors should address this by measuring EdU intensity and cell numbers in early, mid and late S phase. The authors rule out problems with fork progression, but there might be an issue with origin firing.

4. The conclusions regarding RPA based on data in Supplementary Fig. 6a requires further support. The authors should carry out independent biological experiments and address whether the mild changes in RIF1^{-/-} are significant.

It seems that the ssDNA response to HU might be impaired and this could be checked by measuring also RPA intensity in the high RPA population as well as number of cells in several experiments. It would also be interesting to know whether RIF1^{-/-} cells enter mitosis with high levels of ssDNA. This could be addressed by RPA and H3S10P co-staining.

Point-by-point rebuttal

First of all, we thank the reviewers for their constructive comments, which we feel helped us to improve our manuscript. Below, we have provided a point-by-point rebuttal.

Reviewer #1 (Remarks to the Author):

The authors in this paper make the observation that the Rif1 protein is needed to protect stalled replication forks from undergoing extensive, DNA2-dependent, degradation. This correlates with a small defect in fork restart and hypersensitivity to hydroxyurea and cisplatin. The authors also find that the Rif1 interaction with PP1 but not 53BP1 may be important. These observations add Rif1 to the very long (and growing) list of proteins that are needed to prevent fork degradation.

What is not answered here is how Rif1 protects replication forks from nuclease action. The authors suggest it is through PP1, which presumably has many substrates that could be relevant. Fully answering this question may be very difficult. But some simple experiments could be done to try to understand the pathway.

We thank the reviewer for his/her excellent insightful comments. We have now performed multiple experiments to address the issues raised by the referee and believe that this has improved the manuscript and made it much stronger.

There are a large number of experimental and textual issues that need to be addressed:

1. Is the function of Rif1 in fork protection upstream of stabilizing a RAD51 filament on the reversed forks? This is an important question to answer since some but not all fork protection pathways operate upstream of RAD51.

This is indeed a very important issue raised by the referee and we thank him/her for this suggestion. We have now performed experiments where we downregulated RAD51 in both WT and RIF1-deficient cells and assessed fork protection (Fig. 2g) and fork reversal (Fig. 2h). Consistent with earlier published reports that RAD51 is essential for replication fork reversal (Zellweger et al., 2015, Mijic et al., 2017, Bhat et al., 2018) we found that near complete down-regulation of RAD51 in WT and RIF1-deficient MEFs Supplementary fig.3f resulted in no fork degradation in WT cells and a rescue of fork degradation in RIF1-deficient cells (Fig. 2g). Furthermore, our EM experiments show that down regulation of RAD51 resulted in abrogation of fork reversal frequencies in both WT and RIF1-deficient cells (Fig.2h). These data taken together suggest that RAD51 acts in reversing stalled replication forks and these reversed forks are then protected by RIF1 from DNA2-mediated degradation.

2. The iPOND data sets described on page 5 must be included in the manuscript. The heavy/light ratios from each experiment should be reported for all proteins observed in the mass spectrometry experiment. This information is needed to evaluate the claims based on this data.

We have now included the iPOND mass spectrometry data sets with the heavy light ratios for each experiment in (Supplementary table 1) as requested by the reviewer

3. The Bunting lab found 53BP1 to be localized to stalled replication forks by iPOND (Her et al., MCB 2018). Is 53BP1 found in this iPOND dataset? Does it co-localize with Rif1 after HU to EdU foci?

We found enrichment of 53BP1 in only one data set. we used stringent criteria of evaluating factors, and only included factors which were found in both the independent experiments. 53BP1 was therefore not included in our initial analysis. However, we now have included the data from both the experiments in Supplementary table 1.

Does 53BP1 co-localize with Rif1 after HU to EdU foci?

Unfortunately, we could not perform triple co-localization studies on 53BP1, RIF1 and EdU as both the antibodies that work for immunofluorescence analysis were raised in rabbit. However, we performed colocalization studies between EdU and 53BP1 in WT MEFs in the presence or absence of HU (Supplementary fig.1f) Our data show that treatment with cells with HU did not significantly increase the percentage of WT cells with 53BP1-positive EdU foci, which is in contrast to the data we observe for RIF1 recruitment (Fig. 1d and 1e). These data suggest that RIF1 localization to stalled forks is independent of 53BP1. Furthermore, it is interesting to note that the enrichment of 53BP1 at stalled forks by mass spectrometry could represent a subset of forks which have could have collapsed upon HU treatment.

4. The Bunting lab also showed that 53BP1 deficient cells have nascent strand degradation. This is in contrast to what is presented in figure 2d. How many 53bp1^{-/-} cell clones were analyzed in this figure? Is it possible that the cells used picked up a suppressor mutation? Please provide evidence that the 53bp1^{-/-} cells are actually 53bp1-deficient. This is important for the localization studies in Figure 1 as well as the model of fork protection.

We thank the reviewer for pointing out the data from the Bunting lab. We were also intrigued by the contrasting results. Therefore, we carried out fork degradation experiments in additional RIF1 and 53BP1 lines obtained from the lab of Simon Boulton (Chapman et al 2013) (Fig. 2d). Consistent with our earlier data, we did not observe fork degradation in 53BP1-deficient MEFs, whereas RIF1-deficient cells showed fork degradation upon HU treatments. Therefore, the observations of the Bunting lab were not experimentally validated by our own data. Additionally, 53BP1-deficient MEFs did not show an increase in genome instability as observed with metaphase spreads upon HU or cisplatin treatments in contrast to RIF1-deficient cells (Supplementary fig.6b). This is also consistent with the fact that 53BP1-deficient B cells were NOT found to display genome instability upon cisplatin and Mitomycin C treatments (Bunting et al 2012). These data taken together suggest that in our conditions, 53BP1 is not involved in protecting replication forks from degradation. As requested by the reviewer, we also performed western blots in both the 53BP1-deficient cell lines showing that 53BP1 is indeed absent in these cells (Supplementary fig.1b).

5. The Her paper (Her et al., MCB 2018) should be referenced or discussed to put the 53BP1 differences in this manuscript in context with what was previously published.

As suggested by the reviewer, we have now mentioned the discrepancy observed with the Bunting lab in the results section of the manuscript.

6. *Buonomo et al., JCB 2009 showed that Rif1 deficient cells were hypersensitive to replication stress (aphidicolin and MMC). Thus, it is not surprising to find they are also sensitive to HU. This finding should be referenced.*

As suggested by the reviewer, we have now discussed the sensitivity to HU in context with Buonomo et al., in the results section manuscript.

7. *Buonomo et al., also showed that Rif1-deficient cells are defective in HDR. This contrasts with this paper that found no defect in RAD51 foci. The authors should do a more direct measurement of HR repair of DSBs using one of the well-established reporter assays to try to reconcile the different results.*

Indeed, Buonomo et al., 2009, showed defects in HR, ranging from 20-50%. However, multiple later reports (Escribano-Diaz et al., 2013, Isono et al., 2017, Findlay et al., 2018) have shown that down-regulation of RIF1 did not result in defects in HR.

As suggested by the reviewer, we have now performed DR-GFP reporter assays on WT and RIF1-deficient cells. Our data show that there is no significant suppression of HR frequencies in RIF1-deficient cells when compared to WT cells (Supplementary fig.3c). Our data is therefore consistent with the later reports showing that RIF1 is dispensable for HR.

Furthermore, we have also performed reporter based NHEJ assays, which clearly show a decrease in NHEJ frequencies in RIF1-deficient cells, consistent with earlier published data on the role of RIF in NHEJ (Supplementary fig. 6a)

8. *Xu et al., EMBO 2010 reported that Rif1 deficient cells were sensitive to HU. This paper should be referenced.*

We have referenced this paper as suggested by the referee.

9. *Xu et al., ELIFE 2017 further showed that Rif1 and 53BP1 act in the same pathway to maintain viability in HU treated cells. This paper needs to be referenced and discussed since it contrasts with the authors model.*

As requested by the referee, we have now referenced this paper and discussed the data in context of the paper.

10. *The EM data in figure 2f is not convincing. First, the percentage of reversed forks in WT cells is much higher than previously reported by other investigators.*

Indeed, the percentage of reversed forks in our experimental conditions were found to be higher than previously reported for other cells. Most of the experiments done previously on fork reversal frequencies were performed in U2OS cells. We believe that higher percentage of reversed forks observed in our experimental conditions is primarily a cell line specific effect depending on the percentage of S-phase cells and also possibly the number of active forks affected at any given point of time. To test this, we performed FACS analysis for EDU incorporation on U2OS cells and MEFs. Our data shows that the percentage of S-phase cells and also the EdU incorporation intensities are much higher in MEFs compared to the U2OS cells (See figure only for reviewers below). Furthermore, EM analysis on U2OS cells in the presence and absence of HU (See figure only for reviewers below) showed that the percentage of reversed forks observed upon HU treatment was very similar to previously published data. These data taken together, suggest the frequency of fork reversal observed

could be dependent on the percentage of S-phase cells, but importantly, the trend in increase in fork reversal upon replicative stress is conserved across cell types.

Figure1. (a) Cell cycle analysis by EdU incorporation shows the comparative percentage of cells in each phase of cell cycle between WT U2OS and WT MEF cells. (b) Frequency of reversed forks in WT U2OS cells without and with HU treatment. Number of molecules analyzed are indicated within parenthesis. The percentage of reversed forks observed are reported on the second row.

Second it is not clear if there is any difference between the HU-treated WT and *Rif1*^{-/-} cells since there is no measurement of variability. The authors state that similar results were observed in “at least one independent experiment”. Please present those data. Similar concerns need to be addressed for figure 3d.

We apologize to the reviewer for lack of clarity on the variability between EM experiments. We have now included data from 3 independent experiments and added the relevant statistics in the revised manuscript for all EM experiments performed. We have also added a table for all reversed fork measurements across experiments as Supplementary table 3.

11. The authors should show the effect of *PP1i* treatment on fork degradation in *Rif1*^{-/-} cells. The prediction from their model is that *PP1* and *Rif1* should be epistatic.

This is a very important prediction of our model and we thank the reviewer for this suggestion. Indeed, we do observe an epistatic relationship between *PP1* and *RIF1* in the protection of reversed forks. Both siRNA and inhibitor experiments, which either downregulated or inhibited *PP1*, resulted in fork degradation in WT cells upon HU treatments, which were not further exacerbated in *RIF1*-deficient cells. These data suggest that *RIF1* and *PP1* act in the same pathway in the protection of stalled replication forks (Fig. 4e and Supplementary fig. 4d). Furthermore, our EM data also showed correlation with our fiber experiments where inhibition of *PP1* followed by replication stress reduced the fork reversal frequency in WT cells. However, *RIF1*-deficient cells did not show any significant decrease in fork reversal frequencies than what was observed for *RIF1*-deficient cells without *PP1* inhibitor. (Fig. 4f).

12. The authors suggest that *CDK1* phosphorylation of *DNA2* is important for it to degrade forks. This is based on a single experiment (Figure 4f). Much more would need to be done to validate this idea. Does *CDK1i* treatment affect *DNA2* localization to stalled forks? Does it affect fork reversal? What happens when analyzing a *DNA2* mutant that cannot be phosphorylated? Does it affect fork degradation in *BRCA2*^{-/-} cells? Etc.... This should either be removed or further experiments should be done to understand what *CDK1* is doing in this pathway. After all, there may be many *CDK1* targets that are important.

We agree with the reviewer that dissecting the mechanisms involving the role of *CDK1* phosphorylation of *DNA2* would require many more experiments which is beyond the main

focus of this manuscript. We have therefore removed this data from the manuscript as suggested by the reviewer.

We have now strengthened our existing data on PP1 interaction with RIF1 by showing that PP1 is involved in the dephosphorylation of DNA2 upon replication stress. We have performed experiments where DNA2 was immuno-precipitated from nuclear extracts upon HU treatments in the presence or absence of PP1 inhibitor and subsequently probed for the phosphorylation status of DNA2 (Fig. 4e). Our data show that HU treatment in WT cells did not induce an increase in DNA2 phosphorylation levels. However, inhibition of PP1 increased the phosphorylation levels of DNA2 dramatically. Interestingly, RIF1-deficient cells showed high levels of DNA2 phosphorylation upon HU treatments, which were comparable to WT cells treated with HU in the presence of PP1i. Further inhibition of PP1 in RIF1-deficient cells did not increase the DNA2 phosphorylation levels any further, suggesting that RIF1 and PP1 are epistatic in controlling the phosphorylation levels of DNA2 upon replication stress.

13. The results in Figure 6e (top panel) with the DNA2i look to me like it does not rescue the ssDNA region formation effectively. This is in contrast to the complete restoration of fork protection. My interpretation is the opposite of the authors – this data suggests that the fork degradation and generation of ssDNA regions are not linked.

We apologize to the reviewer as the representation of the data was not ideal. We had initially binned the data into different groups and quite a few borderline data points which fell into the highest bin did not give a true picture of the rescue. We have now plotted the ssDNA from each fork as a dot plot and combined all the data points across 3 independent experiments, giving us the opportunity to statistically analyze the differences (Fig. 6e). Our dot plot analysis clearly shows that there is a significant rescue of ssDNA at the forks upon DNA inhibition in RIF1-deficient cells upon restart.

14. The conclusion that protection of reversed forks from degradation “is tightly coupled” to fork restart is an over-interpretation of the data. First, there is only a slight fork restart defect observed and that defect is in the speed of elongation. Second, other fork protection defects (ie BRCA2) do not show any defects in restart. The differences with BRCA2 should be discussed with reference to the Schlacher/Jasin paper originally describing fork protection.

We apologize if the term “tightly coupled” came across as being an over-interpretation of the data. We have made text changes in the manuscript to clarify that our data suggest that protection of fork degradation is linked to a delayed fork restart but not a complete restart defect.

Regarding the lack of restart defects observed in BRCA2-deficient cells observed in Schlacher et al 2011, we believe that the differences observed are due to the different experimental conditions. As the measure of restart, the authors (Schlacher/Jasin) measured the gap (non-incorporation) between two labels (IdU and CldU) in the presence of HU over 5 hours (Fig. 1f of Schlacher/Jasin paper). We believe that this is a readout of fork speed under replicative stress, but not necessarily a robust readout of fork restart. Furthermore, a closer look at the data shows that there are inherent differences between their data in hamster (VC-8) and human BRCA2 (CAPAN-1) mutated cells where the VC-8 cells progress

during HU treatments (showing no difference in the gap) whereas the CAPAN-1 do not (showing differences).

Our experiments, on the other hand, measured the efficiency of restart after the removal of HU, which showed that RIF1-deficient cells have delayed restart upon removal of replicative stress (Fig. 5h and Supplementary fig. 5 c-f). This is in line with data from the Nussenzweig lab (Ray Chaudhuri et al 2016, extended data Fig. 3e) and Vindigni lab (Lemacon et al 2017, Fig. 3f) where they show that loss BRCA2 results in a delayed restart in these cells similar to what we observe in RIF1-deficient cells. Our present work goes a step further to show that that delayed restart could be linked to exposure of ssDNA at and behind the restarted forks resulting in genome instability/DSB formation in late S/G2 phases of the cell cycle.

15. The conclusion that fork restart is defective uses a 1hr recovery time. In these conditions, the authors see a small reduction in length of the label added during the 1h treatment. So restart is actually fine but the authors conclude the restarted forks are slow. However, if inter-origin distance was shorter in the Rif1 mutant cells then it is possible the shorter second label is just because the forks didn't need to move as far to complete replication. A shorter time course would help alleviate this concern.

We thank and appreciate this suggestion from the reviewer. This is a very important question that needs to be addressed. As suggested by the reviewer, we allowed the forks to restart after HU treatment for multiple time points, ranging from 15 mins to 60 mins and measured both the percentage of restarted and stalled forks after HU wash off (Supplementary fig. 5e). We also measured the length of the IdU tracts after release from replication stress (Supplementary fig. 5f). Interestingly, our data show that there was a significant decrease in percentage of restarted forks at early time points after release (15' and 30'). However, at later time points (45' and 60') we did not observe any significant difference in the percentage of restarted forks (Supplementary fig. 5e). These data suggest that RIF1 deficiency results in delayed fork restart after removal of replicative stress which is reflected by the significant decrease of restarted forks in at early time points but not the late time points. This also related with the tract lengths of IdU, where at 15' we do not observe a significant difference. However, the tract lengths at 30', 45' and 1hr show significant differences between WT and RIF1-deficient cells (Supplementary fig. 5f). These data taken together suggest that the differences in track lengths observed between WT and *Rif1*^{-/-} cells are due to a delayed restart and not due to shorter inter-origin distances.

16. A larger point is that the authors make many interesting observations that correlate with each other (ie sensitivity to damage/HU and fork degradation) which could suggest a relationship but they overstate this conclusion. It is also possible that these biological effects of Rif1 inactivation are independent of each other. Considering the large numbers of things that Rif1 has been suggested to do and the pleiotropic functions of PP1, this wouldn't be unexpected. The text and conclusions need to better reflect these uncertainties.

We thank the reviewer for this suggestion, and we have made the necessary changes in the discussion to highlight the multiple roles of RIF1 and PP1 and the possibility that other roles of these proteins could also contribute to the sensitivity.

17. The model and discussion includes the idea that the problem in Rif1 defective cells is due to "aberrant restart through repriming events". There is no evidence presented in the paper

or any experiments completed that addressed “repriming”. The authors do measure ssDNA gaps by EM the origin of those gaps is unclear. I think speculation on this point is fine in the discussion but it should be removed from the model figure and the text needs to either mention other possibilities or make clear that this is speculation.

As suggested by the reviewer, we have removed the re-priming part from the model, and only discussed it as a possibility in the discussion section.

18. The statistical analyses need to be done with more appropriate tests. A Mann-Whitney test is not appropriate when comparing more than 2 samples. A test that takes into account false-discovery rates is needed. This applies to nearly all of the fork protection assays presented since more than 2 samples are compared.

We thank the reviewer for this suggestion. We have performed non-parametric one-way ANNOVA (Kruskal-Wallis test) for experiments where we are comparing more than one group simultaneously for all fiber experiments. Also, a Benjamini-Hochberg post-test was done for taking FDR into account.

19. I don’t understand the point in the discussion about nuclease polarity. Presumably the authors assays are monitoring degradation of both the leading and lagging nascent DNA strands so how could a difference in nuclease polarity be important?

We apologize for the confusion about the discussion of polarity. Indeed, fork degradation takes place on both the strands. We meant to propose that reversed arms with 5’ overhangs could be more susceptible to initial DNA2-mediated degradation, followed by MRE11-mediated degradation on the other strand. However, for the sake for simplicity, we have removed this portion from the discussion section.

20. How long were the cells treated with HU in Figure 1d? Was it less than what would induce DSBs?

The cells were treated for 3 hours with HU. Although we cannot rule out the formation of a few DSBs, these conditions do not induce significant amount of DSBs, as shown by PFGE analysis (Fig. 5a).

21. Page 5: “consistent with earlier published data...” A reference is needed here.

We have added the relevant reference

22. The iPOND data was analyzed with a “two-sided T test”. However, the authors only completed two biological replicates. I don’t think n=2 would provide sufficient statistical power to make a t-test useful.

As suggested by the reviewer, we have now removed the T test and replaced it with the average of fold change between two independent experiments (Fig.1b) and the standard deviation of the selected proteins in Fig. 1c.

23. Page 4 “mechanisms...remains”

We have made the required text change and thank the reviewer for pointing this out.

24. Page 4: “MCM2 deficient mice which are viable...” This is incorrect. The MCM2 mice are not null alleles.

We apologize for this oversight. We have now corrected the mistake and removed the sentence from the introduction.

Reviewer #2 (Remarks to the Author):

Review comments:

This manuscript by Mukherjee et al. describes RIF1 being enriched at stalled replication forks and inhibits DNA2 nuclease activity towards reversed replication forks. This protection of replication forks was dependent on RIF1 interaction with Protein Phosphatase1 and cruciform structures but not RIF1 function in NHEJ. RIF1 deficiency delayed fork restart and increased ssDNA formation and reduced survival in response to replicative stress. Thus, RIF1 play an important role in replication fork protection upon replication stress and therefore contribute to genome stability. Most of the experiments are well designed with proper controls. The data is convincing. Replication stress is believed to be critical for cancer initiation, therefore, stabilization of DNA replication forks is also believed to be essential for maintenance of genome stability and prevention of tumorigenesis. The most widely studied role of RIF1 is in the repair of DSBs via NHEJ, and via its interaction with 53BP1. The hypothesis that a 53BP1 independent function of RIF1 at stalled replication forks, inhibition of DNA2 nuclease activity towards reversed replication forks, provide an unexpected but biologically reasonable potential explanation. Therefore, the proposed research is both biologically and clinically important and interesting.

However, the manuscript by Mukherjee et al. is not written carefully. Multiple figures lack error bars and statistical analysis. The number of repeats and how many fiber/ssDNA/reversed fork/abnormal chromosome were quantified is not mentioned in some cases. In addition, the overall evidence supports most of their claims, but still the authors tend to overinterpret the data. For example, the role of PP1 or CDK1 is really tangential to the story presented, and the connection to RIF1 is unclear.

We thank this reviewer for his/her enthusiasm and that he/she find our experiments “well designed with proper controls” and also that “the data is convincing”. We are also very happy that that reviewer finds the work “both biologically and clinically important and interesting”.

We have now performed additional experiments to address the reviewers’ concerns. Furthermore, and indicated by the reviewer, connection of CDK1 would require much more work and is tangential to the main focus of the manuscript. Therefore, we have removed the section from the manuscript and will be a focus of further studies in our lab. We have however strengthened our observations with PP1 and RIF1 interaction by performing

additional experiments (discussed below) which we believe has improved the manuscript. We have also performed other experiments to address the reviewers' concerns (below).

Here are some specific points:

The major concern:

1. Figure 1d, there is no evidence that these newly synthesized DNA foci are stalled replication forks. Evidence for RIF1 positive foci at stalled replication forks is needed.

We have now performed proximity ligation assay (PLA) with EdU and RIF1 (an imaging base iPOND assay) test if RIF1 is recruited to stalled forks in WT cells. Similar to our immunofluorescence data, our experiments show that although the PLA positive cells are similar in both non-treated and HU treated cells, the number of PLA foci/cell significantly increases upon treatments with HU in these cells. These data have been included as **Supplementary fig. 1d**.

Furthermore, to test if forks are stalled upon HU treatments in our experimental conditions where we observe recruitment of RIF1, we performed the following experiments as described below

- a. Fiber experiments to test of fork progression rates in WT and RIF1-deficient cells in the presence of HU: Our data suggests that there is almost no incorporation of IdU labels upon addition of HU in both WT and RIF1 deficient cells suggesting that the whole population of forks are stalled upon HU treatment. (See below, data only for reviewer)

Figure.2: Tract lengths of CldU and IdU in presence and absence of HU in WT and RIF1 deficient cell lines.

- b. EdU FACS experiment to test the incorporation rates in the presence and absence of HU: Our data show that there is almost no incorporation of EdU in cells treated simultaneously with HU and both WT and RIF1-deficient cells again suggesting that the bulk of the forks are stalled upon treatment with HU (See below, data only for reviewer).

Figure 3: WT and RIF1 deficient cells were labeled with EdU for 3h in presence and absence of HU to compare the incorporation and Intensity of EdU by flowcytometry.

These data taken together suggest that the RIF1 recruitment that we see upon HU treatments could only be at stalled forks and is independent of 53BP1 (Fig1d-e, Supplementary fig 1d and data only for reviewer (above)) whereas RIF1 localization to DSBs is dependent on 53BP1 (Supplementary fig. 1e)

2. Figure 3. Only data suggests involvement of DNA2 as DNA2 inhibitor rescues fork degradation in RIF1 deficient cell. DNA2 is known to drives processing and restart of reversed replication forks in human cells, DNA2 inhibitor can rescue fork degradation in RIF1 deficient cell in many possible ways that is indirect. Evidence needed to demonstrate RIF1 directly interacts with DNA2 in vivo and invitro and inhibit DNA2 nuclease activity in vivo or in vitro.

In addition to DNA2 inhibition (Supplementary fig.4a and Fig.3c), we have now also performed siRNA-based experiments to show that knockdown of DNA2 indeed rescues the fork degradation that is observed in RIF1-deficient cells (Fig. 3a-b), suggesting that the DNA2 inhibitor is specific. Furthermore, consistent with our fiber data, we show that inhibition of DNA2 in RIF1-deficient cells rescues the decrease in reversed fork frequencies in these cells upon replicative stress (fig.3d).

As suggested by the referee, we performed IP experiments to test if RIF1 and DNA2 physically interact to prevent fork degradation. Our IP experiments pulling down DNA2 and probing for RIF1 failed to detect any physical interaction between these proteins both in the presence or absence of HU treatments, suggesting that a physical interaction between RIF1

and DNA2 may not be required to protect replication forks (see below: data only for reviewer). We therefore hypothesized that RIF1 interaction with PP1 could be important for protecting replication forks through de-phosphorylation of DNA2 upon replication stress. To test this, we performed IP experiments where we pulled down DNA2 in the presence or absence of HU and PP1 inhibitor in WT and RIF1-deficient cells. Our data show that in the treatments with HU in WT cells, did not show an increase in DNA2 phosphorylation levels. However, inhibition of PP1 increased the phosphorylation levels of DNA2 dramatically. Interestingly, RIF1-deficient cells showed high levels of DNA2 phosphorylation upon HU treatments which were comparable to WT cells treated with HU in the presence of PP1i. Further inhibition of PP1 in RIF1-deficient cells did not increase the DNA2 phosphorylation levels any further suggesting the RIF1 and PP1 interaction is important for the dephosphorylation of DNA2 upon replicative stress (Fig. 4.g). This is also consistent with our siRNA- and inhibitor-based fiber experiments, where both knockdown and inhibition of PP1 resulted in fork degradation in WT cells. This degradation was not exacerbated in RIF1-deficient cells upon downregulation or inhibition of PP1 (Fig. 4e and Supplementary fig. 4d). Furthermore, these data are consistent with our new EM data, where inhibition of PP1 resulted in decreased fork reversal frequencies in WT cells upon HU treatments, which was also comparable to degradation seen in RIF1-deficient cells. Further inhibition of PP1 in RIF1-deficient cells did not decrease the frequency of fork reversal any further (Fig. 4f). Therefore, we suggest that a direct physical interaction between RIF1 and DNA2 might not be a requirement to prevent uncontrolled fork processing. Fork processing could be mediated through PP1 de-phosphorylation of DNA2 resulting in limited access to forks.

Figure 4: RIF1 does not interact with DNA2. A) Levels of DNA2, RIF1 in nuclear extracts from WT and RIF1^{-/-} MEFs before and after treatment with HU. Western blots were performed with antibody against DNA2 and RIF1 antibody. Histone H3 is used as protein loading control. **B)** Immunoprecipitations were carried out with anti-DNA2 antibody or the corresponding IgG and the immunoprecipates were probed with RIF1 and DNA2 antibody.

3. *Figure 4. The impact of Both PP1 inhibition and CDK1 inhibition are broad and inconclusive. Figure 4e and 4f only provide correlative supporting evidence. CDK1 phosphorylation site on DNA2 is known, phospho-mimetic mutant and phospho-null mutant of DNA2 can provide more direct and specific convincing evidence in both % of reversed forks and fork degradation DNA fiber assay to support conclusion of figure 4e and figure 4f.*

We agree with the reviewer that the CDK1 inhibition experiment was broad and tangential to the main focus of the manuscript. Although the CDK1 phosphorylation sites on yeast Dna2 is known, as per our knowledge there are no documented CDK1 phosphorylation sites on mammalian DNA2. Since this was not the main focus of the manuscript and the identification and characterization of DNA2 phosphorylation would be a line of future studies in the lab, we decided to take the experiment out of the present manuscript (as also suggested by reviewer1).

However, we have strengthened our data on the PP1 interaction with RIF1 and its role on modulating the phosphorylation levels of DNA2 (See response to point 2), which we believe could be important for the protection of fork degradation.

4. *Figure 6a, Increased DSB in RIF1-/- can be due to defect in DSB repair during recovering from HU induced replication stress. Evidence needed to demonstrate the increased DSB is not all due to decreased DSB repair. Supplementary figure 3a 3b only demonstrate change in HR is not significantly affected, evidence showing that NHEJ is not affected is needed.*

We thank the reviewer for this suggestion. We have now used a reporter assay to assess the efficiency of NHEJ (Certo et al 2011) in WT and RIF1-deficient cells. Consistent with previously published evidence (Chapman et al 2013), RIF1-deficient cells showed a significant decrease in NHEJ efficiency (Supplementary fig. 6a).

We cannot formally rule out that part of the increased DSBs after release from HU is due to unrepaired DSBs through dysfunctional NHEJ. However, we have also performed genome instability analysis in WT and 53BP1-deficient cells. In contrast to RIF1 deficiency, loss of 53BP1 did not result in significant increase of chromosomal aberrations upon both HU and cisplatin-mediated replication stress (Supplementary fig. 6b). These data suggest that the genome instability observed in RIF1-deficient cells is not mediated through loss of NHEJ. We have discussed this possibility of the role of dysfunctional NHEJ in DSBs in the revised discussion section of the manuscript.

5. *Supplemental Figure 6, interpret RPA staining as marker for ssDNA in flow cytometry experiment is difficult because only chromatin bound RPA can be used to evaluate the amount of ssDNA. The RPA result also contradict the EM results. EM result in figure 6e shows huge increase in ssDNA 30 minutes after release from HU, while RPA results shows similar ssDNA 5 hours after release and increase after 15 hours release.*

We have indeed stained for chromatin-bound RPA for FACS analysis. To detect only chromatin-bound RPA, we subjected all the samples to pre-extraction prior to fixation after treatments.

We apologize if the interpretation of the data was not clear. The RPA experiment was performed to follow retention of ssDNA in WT and RIF1-deficient cells over a longer period

of time after removal of replicative stress. The similar amounts of RPA in WT and RIF1 could be due to the fact that only a fraction of the forks has restarted and the RPA signals are coming from stalled forks. It would be technically difficult to distinguish changes in ssDNA of few 100 nucleotides through a bulk assay like FACS analysis. However, the increase in ssDNA in RIF1-deficient cells became evident at 15 hrs, when most of the cells had completely restarted their forks. The increase of under-replicated DNA becomes clear at these time points, since the cells have accumulated enough ssDNA in RIF1-deficient cells but not WT. The EM experiments, however, provided us with a “snap shot” of the early events in fork restart, where subtle changes are visible when only a fraction of the stalled forks are restarted. This gives us insights into the early events during restart of the replication forks, which in RIF1-deficient conditions showed increased amount of ssDNA at and behind the forks. We have changed our wording in the revised manuscript in order to clarify this point.

Minor concerns:

1. Provide table with list of top hits with p-value FDR etc. for figure 1b.

We have now included the whole data set with relevant statistics in Supplementary table 1

2. Figure 2f, no error bar not statistics analysis. no number of repeats

Statistics and number of analyzed molecules for EM experiments are now included in Fig. 2f, while detailed scoring from each experiment is included in Supplementary table 3a.

3. Figure 3d, no error bar not statistics analysis. no number of repeats

Statistics and number of analyzed molecules are now included in Fig. 3d, while detailed scoring from each experiment is included in Supplementary table 3c

4. Figure 4d, no error bar not statistics analysis. no number of repeats

Statistics and number of analyzed molecules are now included in Fig. 4d, while detailed scoring from each experiment is included in Supplementary table 3d.

5. Figure 6a, quantification and represented image are very contradictory. The image shows IR have much higher DSB than rif1-/- 15hr release but quantification suggest otherwise. Also, WT HU have higher DSB than WT 15 hrs release, but quantification suggest otherwise. Quality of representative image is low and longer exposure may better demonstrate the result.

We thank the reviewer for this suggestion. We have now provided an image with higher exposure to bring out the differences between samples (Fig 6a). The quantifications are done across the whole lane of the sample (the main band with broken DNA and also the low molecular weight DNA which appears as a smear) and normalized against the loading (unbroken band in the wells). Therefore, an increase of the smear adds to the amount of

DSBs quantified. Both WT and Rif1-deficient cells upon release show higher amounts of low molecular weight DNA, thus representing higher amounts of DSB formation when compared to its counterparts at earlier time points.

6. *Figure 6e, no error bar, no statistics analysis, no number of repeats or number of molecules were quantified.*

As requested, we have added the statistics to the figure which have now been labelled as **Fig. 6e and 6f**. The number of repetitions and the total number of molecules analyzed are now a part of the figure legends.

7. *The quality of WB in supplementary figure 2a and 2c is poor and needs to be repeated.*

We have replaced the western blots with better bots. They are now **Supplementary fig. 2a and 2e**.

Reviewer #3 (Remarks to the Author):

The manuscript by Mukherjee and colleges describes a role for RIF2 in preventing degradation of reversed replication fork that involves its ability to bind PP1 but is independent of 53BP1. Rif1 specifically prevents DNA2 from attacking reversed forks to allow efficient restart. This is an interesting new role for RIF1 that should be of interest to the genome maintenance and replication fields. Overall, the authors present convincing evidence to support their conclusions and, given that the critical points below concerning data availability and statistics are addressed, I recommend this work for publication.

We are very happy that the reviewer finds this study to be “of interest to the of genome maintenance and replication fields” and thank the reviewer for his/her enthusiasm for publication of this manuscript. We have extensively addressed the comments of the reviewers which can be found below.

Concerns

1. *I could not find the results of the mass spec experiments in the manuscript. This is an important resource from this work and must be included to allow comparison of their iPOND data to other published data sets and validate the quality of the experiments. The full list of proteins with the SILAC ratios should be provides as a table and the raw ms data should be deposited in a proteome data repository according to best practice in the field.*

We have now included the datasets from the iPOND experiments as a table in **Supplementary table 1** iPOND. Furthermore, the datasets will be uploaded to a relevant repository if the manuscript is accepted.

2. In many experiments it is unclear whether statistics includes biological variation between independent experiments or merely variation between observations in one experiments.

This should be clarified, and as a minimum the authors should confirm that two biological independent experiments showed similar results for:

Fig. 2a, b, c, d

Fig. 3b, c

Fig. 4 c, e, f

Fig. 5h

Fig. 6 f,g

We thank the reviewer for this comment. The variation between observations shown in the fiber experiments are from a single representative experiment. We have now added tables for the fiber experiments mentioned above, including three biological replicates with the relevant statistics in Supplementary table 2.

For the EM experiments, statistics are missing for the following experiments and it is unclear how many biological replicates they represent

Fig. 2e

Fig. 3d

Fig. 4d

Fig. 6e

We have now added the relevant statistics to the EM experiments and each experiment has now been repeated three times. We have also included a table showing the individual data from each biological replicate (Supplementary table 3).

3. In the EdU facs analysis supplementary fig. 2a there appears to be more S phase cells with low EdU incorporation in the absence of RIF1 and the distribution of cells within S phase also appears to be skewed towards early S compared to wt cells. This might reflect changes in replication timing or that a higher fraction of cells experiences problems during normal replication. The authors should address this by measuring EdU intensity and cell numbers in early, mid and late S phase. The authors rule out problems with fork progression, but there might be an issue with origin firing.

We thank the reviewer for this comment. We have now looked into the FACS profiles of RIF1-deficient cells in detail. As suggested by the referee, we measured the percentage of cell in early medium and late S-phase. Although there is a trend of higher number of cells in early S-phase and a slightly lower percentage in late S-phase in RIF1-deficient cells, these differences turned out not to be statistically significant across experiments. Furthermore, as suggested by the referee, we also compared the high and low EdU intensities in WT vs RIF1 cells. However, we could not find any statistically relevant differences across 3 independent experiments (Supplementary fig. 2a- e).

4. The conclusions regarding RPA based on data in Supplementary Fig. 6a requires further support. The authors should carry out independent biological experiments and address whether the mild changes in RIF1-/- are significant.

It seems that the ssDNA response to HU might be impaired and this could be checked by measuring also RPA intensity in the high RPA population as well as number of cells in several

experiments. It would also be interesting to know whether RIF1^{-/-} cells enter mitosis with high levels of ssDNA. This could be addressed by RPA and H3S10P co-staining.

This is indeed a very important point. We thank the reviewer for this suggestion. We have analyzed the RPA signals across 3 independent biological replicates. We do indeed see a statistically significant increase in percentage of RPA-positive cells at 15 hours after release from HU-mediated replicative stress in RIF1-deficient cells (Supplementary fig. 6c-e). Further dissection of the accumulation of RPA intensities revealed that cells with highest RPA signals upon RIF1 deficiency were significantly increased in late S-G2 phases of the cell cycle at 15 hours after HU release, suggesting the presence of under-replicated region upon release from replicative stress.

Finally, as per the suggestion of the referee, we tested if RIF1-deficient cells enter mitosis with higher levels of RPA2, using phospho-serine-10 HistoneH3 co-staining. Quantitation of phospho-HistoneH3-positive cells did not reveal a significant increase of mitotic cells in RIF1-deficient cells, and high levels of RPA2 did not coincide with phospho-HistoneH3 staining at 15 hours after HU release. These results suggest that the accumulation of RPA is restricted to late S-G2 cells (Supplementary fig. 6f).

Reviewers' comments:

Reviewer #1 (Remarks to the Author):

The authors have done a considerable amount of new work to better validate their conclusions. These conclusions are interesting especially since they contrast with some previously published findings. Thus, the paper will certainly impact the field and change some of the current thinking. I have only a few minor comments that the authors should consider.

1. The authors' response to comment 1 did not answer my question. The question asked was not whether fork reversal is required for degradation. Rather, I was asking whether RIF1 acts via RAD51 to block degradation? In other words, is RAD51 filament stability on the reversed fork regulated by RIF1? I'm assuming this is the author's model but direct experimental confirmation of this point would be useful. For example, they could overexpress RAD51 to determine if that rescues the Rif1-deficient phenotype.

2. The iPOND data description from the authors is that 44 proteins had greater than 2-fold enrichment. Is this based on the average of two experiments? If so, the average should be shown in the supplementary data table and it would improve clarity if these 44 proteins were highlighted in some way.

Reviewer #2 (Remarks to the Author):

The authors have addressed most of the issues I raised.

Reviewer #3 (Remarks to the Author):

The authors have addressed most of my comments adequately with one exception (see below). Once this is addressed, I recommend the manuscript for publication.

In response to the point regarding biological replicates of their fiber data, the authors have provided information on 3 biological replicates in a supplementary table. This is important and a major improvement. However, as the authors have 3 biological replicates of their fiber experiments, they should make these data easily available to the reader. In the main figures they show a single experiment with statistics on the variation between measurements (largely irrelevant). The relevant information for the reader is the variation between biological replicates. Preferably they would show the mean of 3 biological replicates with stats in the main figures (data now in supplementary table) or as a minimum this has to be in supplements as a figure with stats clearly visible and referred to in each figure legend (not as a table).

Rebuttal

Reviewer #1 (Remarks to the Author):

The authors have done a considerable amount of new work to better validate their conclusions. These conclusions are interesting especially since they contrast with some previously published findings. Thus, the paper will certainly impact the field and change some of the current thinking. I have only a few minor comments that the authors should consider.

We thank the reviewer for the positive feedback and are very happy that the reviewer thinks that the paper will impact and change some of the current thinking in the field.

We have added additional data to the manuscript in support of the hypothesis that RIF1 could stabilize RAD51 at reversed forks and also discussed the technical challenges to performing overexpression of RAD51 below.

1. *The authors' response to comment 1 did not answer my question. The question asked was not whether fork reversal is required for degradation. Rather, I was asking whether RIF1 acts via RAD51 to block degradation? In other words, is RAD51 filament stability on the reversed fork regulated by RIF1? I'm assuming this is the author's model but direct experimental confirmation of this point would be useful. For example, they could overexpress RAD51 to determine if that rescues the Rif1-deficient phenotype.*

We apologize if we did not get the question correctly in the first place. Our data would suggest that in addition to controlling the access of DNA2 to reversed forks (through its interaction with PP1), RIF1 could also act by regulating RAD51 stability on the reversed arm.

We have generated fiber data where RAD51 downregulation was not complete (80%) (Figure1, below and Supplementary Figure 3g and h in the manuscript). As demonstrated earlier (Bhat et al., 2018) incomplete downregulation of RAD51 in WT MEFs induced fork degradation in these cells suggesting that the remaining amount of RAD51 although sufficient to mediate fork reversal is not enough to confer fork protection. Furthermore, partial down regulation of RAD51 in RIF1-deficient cells did not rescue the fork degradation these cells and also did not show a significant additive effect on degradation. This data suggests that effect RAD51 and RIF1 could be epistatic in stabilizing the reversed arm and RIF1 could be involved in stabilization of RAD51 at reversed forks.

Figure1. (left) Western blot analysis for the down-regulation of RAD51 in WT and RIF1-deficient MEFs with 50nmol siRNA which results in approximately 80% down regulation of the protein.

(right) DNA fiber analysis of WT and RIF1-deficient MEFs transfected with either mock or siRAD51 and treated with HU

Regarding the reviewers' suggestion on overexpressing RAD51 to assess rescue of RIF1 degradation phenotype, we believe that interpretation of the data could be hampered due to the below mentioned technicalities.

1. We do not think that that availability of RAD51 is an issue upon RIF1 deficiency. As shown above (Figure1) RIF1-deficient cells have similar levels of RAD51 when compared to WT cells.

This suggests that loading of RAD51 to the reversed arm is the limiting step which would require mediators for RAD51 loading (BRCA1/2, BOD1L1, RIF1 etc).

2. We have preliminary data from live cell imaging upon RAD51 overexpression in mouse cells. RAD51 overexpression results in deposition of the protein in between chromosomal territories and also results in massive cytoplasmic accumulation (Figure2, below). Therefore, we believe performing an experiment in such conditions would result in difficulties with data interpretation. Furthermore, overexpression of RAD51 has also been previously reported to promote genome instability and aneuploidy (Richardson et al., 2004, Oncogene) probably due to non-specific binding to DNA and promoting unwanted recombination events.

Therefore, we have addressed this issue textually where we discussed the possibility that RIF1 could also be involved in the stabilization of RAD51 filaments on the reversed arm thus preventing degradation.

Figure2. Live cell imaging for GFP-RAD51 in WT MEFs. White arrows indicate accumulation of RAD51 either at spaces between chromosome territories or in the cytoplasm.

The iPOND data description from the authors is that 44 proteins had greater than 2-fold enrichment. Is this based on the average of two experiments? If so, the average should be shown in the supplementary data table and it would improve clarity if these 44 proteins were highlighted in some way.

As suggested by the reviewer we have added the 44 proteins as a separate tab in the supplementary table with the average values between 2 experiments.

Reviewer #2 (Remarks to the Author):

The authors have addressed most of the issues I raised.

We thank this reviewer for his/her enthusiasm and support for the publication of the manuscript

Reviewer #3 (Remarks to the Author):

The authors have addressed most of my comments adequately with one exception (see below). Once this is addressed, I recommend the manuscript for publication.

In response to the point regarding biological replicates of their fiber data, the authors have provided

information on 3 biological replicates in a supplementary table. This important and a major improvement. However, as the authors have 3 biological replicates of their fiber experiments, they should make these data easily available to the reader. In the main figures they show a single experiment with statistics on the variation between measurements (largely irrelevant). The relevant information for the reader is the variation between biological replicates. Preferably they would show the mean of 3 biological replicates with stats in the main figures (data now in supplementary table) or as a minimum this has to be in supplements as a figure with stats clearly visible and referred to in each figure legend (not as a table).

We thank this reviewer for his/her support and the recommendation for the publication of the manuscript.

As suggested by this referee, we have now added the mean and SEM from each fiber experiment replicates in **Supplementary Figure 7 and 8** and also mentioned it in the text of the manuscript.